# The mortality cost of carbon

R. Daniel Bressler [1,2,3✉]

Many studies project that climate change can cause a significant number of excess deaths. Yet, in integrated assessment models (IAMs) that determine the social cost of carbon (SCC) and prescribe optimal climate policy, human mortality impacts are limited and not updated to the latest scientific understanding. This study extends the DICE-2016 IAM to explicitly include temperature-related mortality impacts by estimating a climate-mortality damage function. We introduce a metric, the mortality cost of carbon (MCC), that estimates the number of deaths caused by the emissions of one additional metric ton of $CO_2$. In the baseline emissions scenario, the 2020 MCC is $2.26 \times 10^{-4}$ [low to high estimate $-1.71 \times 10^{-4}$ to $6.78 \times 10^{-4}$] excess deaths per metric ton of 2020 emissions. This implies that adding 4,434 metric tons of carbon dioxide in 2020—equivalent to the lifetime emissions of 3.5 average Americans—causes one excess death globally in expectation between 2020-2100. Incorporating mortality costs increases the 2020 SCC from \$37 to \$258 [−\$69 to \$545] per metric ton in the baseline emissions scenario. Optimal climate policy changes from gradual emissions reductions starting in 2050 to full decarbonization by 2050 when mortality is considered.

[1] Columbia University School of International and Public Affairs, New York, NY, USA. [2] The Earth Institute at Columbia University, New York, NY, USA. [3] Columbia University Center for Environmental Economics and Policy, New York, NY, USA. ✉email: rdb2148@columbia.edu

The social cost of carbon (SCC) is arguably the single most important concept in the economics of climate change[1]. It represents the marginal social damage from emitting one metric ton of carbon dioxide-equivalent at a certain point in time[2]. According to standard economic theory, it represents the price that should be put on carbon dioxide to reduce emissions to socially optimal levels along the optimal emissions trajectory[3]. The SCC has been highly influential in informing climate policy. For example, regulations with benefits totaling over $1 trillion in the United States have used the SCC in their economic analysis[1]. The SCC is commonly estimated using climate-economy integrated assessment models (IAMs), which synthesize the state of scientific knowledge to inform policy[4,5]. Climate-economy IAMs that produce an SCC also project the optimal path of future emissions by comparing climate damages with the cost of reducing emissions.

Despite the theoretical and policy importance of the SCC, many commentaries have argued that current estimates of the SCC remain inadequate[5–12]. One major line of criticism is that IAMs do not represent the latest scientific understanding of climate impacts. Although substantial advances in climate impact research have been made in recent years, IAMs are still omitting a significant portion of likely damages[13,14]. Another major line of criticism is that a wide variety of climate damages—sea level rise, extreme weather, the direct effects of heat on productivity, agricultural impacts, and many more—must be monetized and summarized into a single number, and the relative contribution of these damages is often unclear[11,13,15]. In addition, the magnitude of climate damages is sensitive to subjective choices around the monetization of non-market damages, and, since damages occur over long timescales, the discount rate at which future damage is converted into present value[5,10,11,15].

One source of climate damages not updated to the latest scientific understanding in IAMs is the effect of climate change on human mortality. A 2017 National Academy of Sciences report specifically mentioned mortality as a damage source that could be immediately updated in IAMs[5]. A large body of literature suggests that climate change is likely to have a significant effect on temperature-related mortality[16–56]. A *Lancet* report concluded that "Climate change is the biggest global health threat of the 21st century"[16]. Yet, climate-mortality damages are currently limited in the most widely used IAMs. In FUND, mortality costs account for ~3% of total damages[13]. In DICE-2016, mortality impacts are not updated to the latest scientific understanding and less than 5% of the damages come from mortality (see "Methods" and Supplementary Materials for details).

In this study, we create an extension to DICE-2016 called DICE-EMR (Dynamic Integrated Climate-Economy Model with an Endogenous Mortality Response). We construct an additional reduced-form mortality damage function that projects the effect of climate change on the mortality rate using estimates from studies chosen from an interdisciplinary systematic research synthesis of the scholarly literature (see "Methods" section for details). We use DICE-EMR to produce a new metric that avoids some of the limitations of the SCC: the mortality cost of carbon (MCC). The 2020 MCC is the number of expected temperature-related excess deaths globally from 2020 to 2100 caused by the emission of one additional metric ton of carbon-dioxide-equivalent emissions in 2020. We find that in the DICE baseline scenario that results in 4.1 °C warming above preindustrial temperatures by 2100, the 2020 MCC is $2.26 \times 10^{-4}$ lives per metric ton in the central estimate, which implies that adding 4,434 metric tons of carbon dioxide in 2020—equivalent to the lifetime emissions of 3.5 average Americans—causes one excess death globally in expectation between 2020 and 2100. We also update the SCC and the optimal climate policy from DICE-2016 in DICE-EMR. After incorporating mortality costs in DICE-EMR, the 2020 SCC increases over sevenfold to $258 per metric ton in the central estimate in the baseline emissions scenario, and optimal climate policy changes from gradual emissions reductions starting in 2050 to full decarbonization by 2050.

## Results

**The MCC.** The 2020 MCC is the number of expected temperature-related excess deaths globally from 2020 to 2100 caused by the emission of one additional metric ton of carbon-dioxide-equivalent emissions in 2020 (see "Methods" section for technical details). Excess deaths are deaths attributable to climate change that occur prematurely relative to a counterfactual scenario in which the marginal emission did not occur. To provide further resolution into the mortality effect of marginal emissions over time, the MCC can be disaggregated across years, an exercise we do in the "Discussion" section. The SCC is similar to the MCC in that both metrics quantify the damage from a marginal increase in emissions in a certain year. The main differences between the SCC and the MCC are: (1) the SCC is intended to include all market and non-market damages from marginal emissions whereas the MCC only measures the effect of marginal emissions on excess deaths; (2) the SCC monetizes all climate damages into a single consumption-equivalent value whereas the MCC does not monetize damages because it is in units of excess deaths; and (3) the SCC converts future damages to present value through discounting whereas the MCC is simply the number of excess deaths from 2020 to 2100. Discounting and valuing lives is a complex and controversial issue. The MCC provides a measure of the mortality damage from marginal emissions without discounting or valuing lives. For these reasons, the MCC is a more straightforward and transparent estimate of the marginal effect of carbon emissions compared to the SCC.

Like the SCC, the MCC is useful for determining the social impact of new marginal activities or projects that produce greenhouse emissions, or, equivalently, the benefit from forgoing these activities. In the central estimate, we find that the 2020 MCC is $2.26 \times 10^{-4}$ lives per metric ton as shown in Table 1, with the low (<10th percentile) to high (>90th percentile) estimate ranging from $-1.71 \times 10^{-4}$ to $6.78 \times 10^{-4}$ (see the "Methods"

**Table 1 2020 mortality cost of carbon (MCC).**

|  | Low mortality estimate (<10th percentile) | Central mortality estimate | High mortality estimate (>90th percentile) |
|---|---|---|---|
| Baseline emissions scenario (4.1 °C warming by 2100) | $-1.71 \times 10^{-4}$ | $2.26 \times 10^{-4}$ | $6.78 \times 10^{-4}$ |
| Optimal emissions scenario (2.4 °C warming by 2100) | $-2.16 \times 10^{-4}$ | $1.07 \times 10^{-4}$ | $5.22 \times 10^{-4}$ |

DICE-EMR projects that an additional metric ton of carbon dioxide emitted in 2020 causes $2.26 \times 10^{-4}$ excess deaths from 2020 to 2100 in the central estimate in the baseline emissions scenario and $1.07 \times 10^{-4}$ excess deaths in the central estimate in the optimal emissions scenario.

**A**

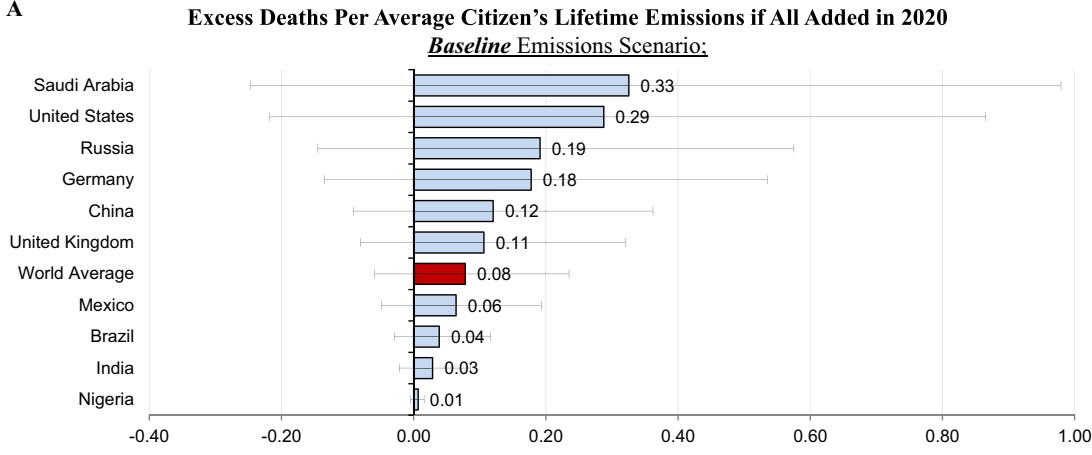

**Excess Deaths Per Average Citizen's Lifetime Emissions if All Added in 2020**
*_Baseline_ Emissions Scenario;*

**B**

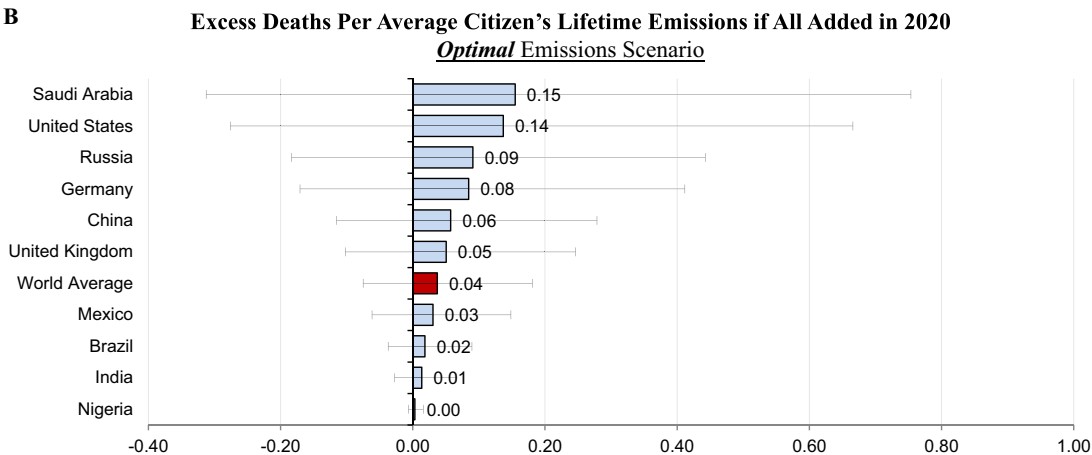

**Excess Deaths Per Average Citizen's Lifetime Emissions if All Added in 2020**
*_Optimal_ Emissions Scenario*

**Fig. 1 Implications of the 2020 mortality cost of carbon.** Average lifetime emissions are calculated as 2017 carbon dioxide emissions production per capita multiplied by 2017 life expectancy at birth. The error bars show the low (<10th percentile) and high (>90th percentile) mortality estimates (see "Methods" section for more details on uncertainty). **A** The 2020 MCC in the baseline emissions scenario is $2.26 \times 10^{-4}$ excess deaths per metric ton of 2020 emissions in the central estimate. This implies that the lifetime emissions of an average American (1,276 metric tons) causes 0.29 excess deaths in expectation if all added in 2020, the lifetime emissions of an average Indian (127 metric tons) causes 0.03 excess deaths in expectation if all added in 2020, and the lifetime emissions of an average person in the world (347 metric tons) causes 0.08 excess deaths if all added in 2020. **B** The 2020 MCC in the optimal emissions scenario is $1.07 \times 10^{-4}$ excess deaths per metric ton of 2020 emissions in the central estimate. This implies that the lifetime emissions of an average American (1,276 metric tons) causes 0.15 excess deaths in expectation if all added in 2020, the lifetime emissions of an average Indian (127 metric tons) causes 0.01 excess deaths in expectation if all added in 2020, and the lifetime emissions of an average person in the world (347 metric tons) causes 0.04 excess deaths if all added in 2020.

section for more details on the uncertainty in the mortality estimate). This implies that reducing (increasing) emissions by 1 million metric tons of carbon dioxide in 2020 saves 226 lives (causes 226 excess deaths) in expectation from 2020 to 2100 in the baseline emissions scenario. One million metric tons is roughly equal to the average annual emissions of 35 commercial airliners, 216,000 passenger vehicles, and 115,000 homes in the United States. The MCC implies that removing (adding) a year's worth of carbon dioxide emissions from an average coal-fired powerplant in the United States in 2020 saves 904 lives (causes 904 excess deaths) in expectation from 2020 to 2100 (refs. [57,58]).

Our central estimate 2020 MCC also implies that reducing (adding) 4,434 metric tons of carbon dioxide in 2020 saves one life (causes one excess death) in expectation globally between 2020 and 2100. In all, 4,434 metric tons is equivalent to the lifetime emissions of 3.5 average Americans, 146.2 Nigerians, and 12.8 average world people. Figure 1A shows the reciprocal of these country-level figures: the average number of lives saved (excess deaths caused) by reducing (increasing) emissions in 2020

by the metric tonnage equal to an average citizen's lifetime emissions. As the figure shows, we find that adding (reducing) 1,276 metric tons of carbon dioxide in 2020—equivalent to the lifetime emissions of an average American—causes (reduces) 0.29 excess deaths between 2020 and 2100 in expectation. Adding (reducing) 127 metric tons of carbon dioxide in 2020—equivalent to the lifetime emissions of an average Indian—causes (reduces) 0.03 excess deaths between 2020 and 2100 in expectation. Adding (reducing) 347 metric tons of carbon dioxide in 2020—equivalent to the lifetime emissions of an average person in the world—causes (reduces) 0.08 excess deaths between 2020 and 2100 in expectation.

**The SCC and optimal climate policy.** Although the MCC is a useful and more transparent metric for determining the mortality consequences of marginal emissions choices, the SCC is still necessary for cost–benefit analysis and for determining the optimal price on carbon. In addition to the MCC, we also use DICE-EMR to estimate the SCC as well as the optimal climate

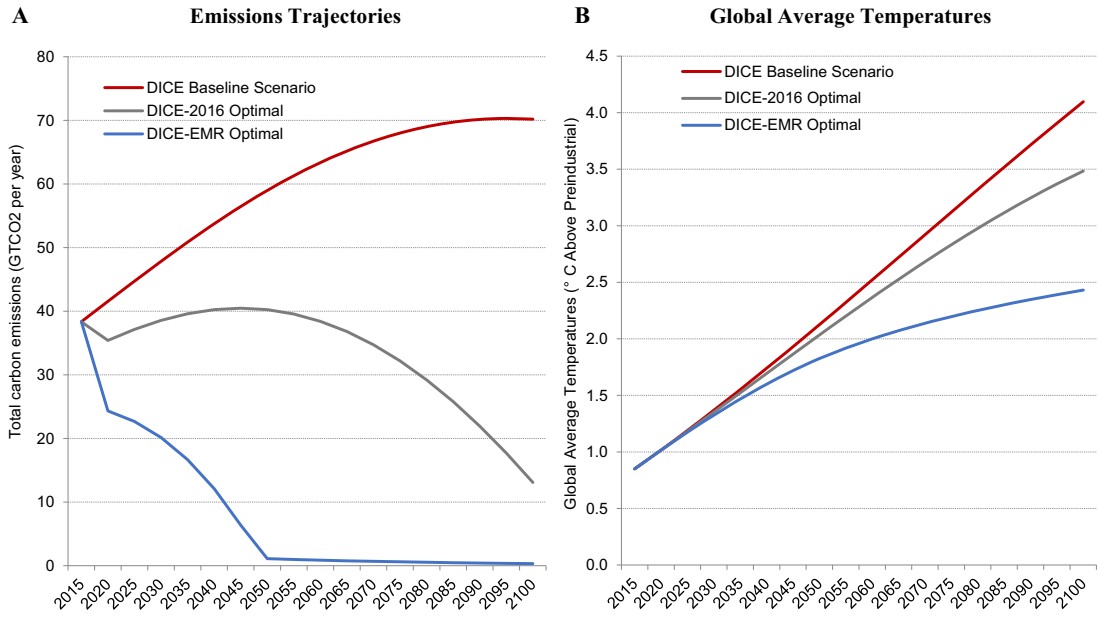

**Fig. 2 Optimal Climate Policy.** Integrated assessment models (IAMs) assess the cost of reducing emissions and the damages from climate change in a dynamic general equilibrium model that includes coupled interactions between the economy and the climate. They can be used normatively to determine optimal climate policy. They do this by using optimal control to determine a path of emissions trajectories that optimizes the net present value of social welfare. **A** Optimal climate policy in DICE-2016 involves gradual emissions reductions starting in 2050 while optimal climate policy in DICE-EMR involves immediate emissions reductions and full decarbonization by 2050. **B** Optimal climate policy in DICE-2016 results in 3.5 °C warming by 2100 while optimal climate policy in DICE-EMR results in 2.4 °C warming by 2100.

policy. The SCC in DICE-EMR includes two sources of climate damages: (1) climate damages to economic output from the original DICE climate-economy damage function, which we retain, and (2) the consumption-equivalent welfare loss from the mortality consequences of climate change (see "Methods" section for details). Besides adding the mortality damage function and incorporating the subsequent welfare loss and feedbacks, DICE-EMR adopts all other structure, equations, base parameters, including discount rates, the original climate-economy damage function, and the baseline emissions scenario of the DICE-2016 model in order to isolate the effect of accounting for temperature-related mortality in DICE. Excess deaths do not need to be monetized and discounted to estimate the MCC, but they do need to be monetized and discounted to estimate the SCC and optimal emissions trajectories. To do this, we calibrate the consumption-equivalent welfare loss from higher mortality in a representative agent general equilibrium model with endogenous mortality (see "Methods" section for details).

Because temperature-related mortality projections are highly convex in global average temperatures, i.e., mortality increases at an increasing rate in global average temperature (see "Discussion" section), societies have a strong incentive to avoid scenarios where global average temperatures are especially damaging. This causes a large difference in optimal climate policy in DICE-EMR compared to DICE-2016 (see Fig. 2). Optimal climate policy in DICE-2016 involves an emissions plateau and then gradual reductions starting in 2050. This results in 3.5 °C warming by 2100. We find that optimal climate policy in DICE-EMR, however, involves large immediate emissions reductions and full decarbonization by 2050. This results in 2.4 °C warming by 2100.

If the world undertakes the optimal emissions path in DICE-EMR and restrains global average temperatures to 2.4 °C, we largely avoid the temperatures where marginal increases in temperature resulting from a marginal emission today are most

damaging. Therefore, the SCC and the MCC are highly sensitive to future climate policy. On the optimal emissions path, the 2020 SCC drops by 39% from $258 in the baseline emissions scenario to $158 per metric ton (see Table 2) and the 2020 MCC drops by 53% from $2.26 \times 10^{-4}$ lives per metric ton in the baseline emissions scenario to $1.07 \times 10^{-4}$ lives per metric ton (see Table 1). This implies that under DICE-EMR's optimal climate policy, adding (reducing) 9,318 tons of carbon dioxide—equivalent to the lifetime emissions of 7.3 average Americans—causes (reduces) one excess death globally between 2020 and 2100. It also implies that adding (reducing) 1,276 metric tons of carbon dioxide in 2020—equivalent to the lifetime emissions of an average American—causes (reduces) 0.14 excess deaths between 2020 and 2100 in expectation on the optimal emissions path (see Fig. 1B).

In addition to marginal effects, we find that pursuing the DICE-EMR optimal emissions path has significant mortality benefits in aggregate over the twenty-first century. Pursuing the DICE-EMR optimal emissions path saves a projected 74 million lives over the course of the twenty-first century (see Fig. 3), as the number of temperature-related excess deaths falls from 83 million in the DICE baseline emissions scenario to 9 million in the DICE-EMR optimal emissions scenario.

## Discussion

In this paper, we introduced a metric: the MCC. This metric is useful for calculating the marginal mortality effects of emissions. We have shown that in the DICE baseline emissions scenario that results in 4.1 °C warming by 2100, the central estimate MCC is significant. It implies that adding 4,434 metric tons of carbon dioxide in 2020—equivalent to the average lifetime emissions of 12.8 average world people or 3.5 Americans—causes one excess death globally in expectation between 2020 and 2100.

| Table 2 2020 social cost of carbon (SCC). | | | | | |
|---|---|---|---|---|---|
| | VSLY = per capita consumption | VSLY = 2× per capita consumption | VSLY = 4× per capita consumption | VSLY = 8× per capita consumption | VSLY = 16× per capita consumption |
| **DICE baseline emissions scenario** | | | | | |
| 2020 SCC | $135 | $177 | $258 | $414 | $789 |
| Mortality response uncertainty | [−$6, $255] | [−$28, $349] | [−$69, $545] | [−$140, $915] | [−$274, $1621] |
| **Optimized emissions scenario** | | | | | |
| 2020 SCC | $91 | $110 | $158 | $221 | $430 |
| Mortality response uncertainty | [−$74, $233] | [−$77, $309] | [−$142, $475] | [−$267, $816] | [−$556, $1,333] |

In the primary specification, DICE-EMR projects that the 2020 social cost of carbon is $258 in the baseline emissions scenario and $158 in the optimized emissions scenario. These figures vary with the value of statistical life year (VSLY) estimates.

**Fig. 3 Cumulative number of excess deaths from climate change in DICE-EMR.** The DICE baseline emissions scenario results in 83 million cumulative excess deaths by 2100 in the central estimate. Seventy-four million of these deaths can be averted by pursuing the DICE-EMR optimal emissions path, which results in 2.4 °C of warming and nine million deaths by the end of the century. The red shaded region shows the range between the high (>90th percentile) and low (<10th percentile) projections for excess deaths in the DICE baseline emissions scenario. The blue shaded region shows the range between high and low projections for excess deaths in the DICE-EMR optimal emissions path.

In total, we find that there are 83 million projected cumulative excess deaths between 2020 and 2100 in the central estimate in the DICE baseline emissions scenario. By the end of the century, the projected 4.6 million excess yearly deaths would put climate change 6th on the 2017 Global Burden of Disease risk factor risk list ahead of outdoor air pollution (3.4 million yearly excess deaths) and just below obesity (4.7 million yearly excess deaths)[59,60].

This large marginal effect may seem counterintuitive compared to the relatively more modest aggregate effect shown in Fig. 4A. However, just considering the total effect belies the significant impact that marginal emissions decisions today have on mortality over the twenty-first century. What matters for the impact of marginal emissions is not the aggregate number of deaths, but the first derivative of the mortality damage curve, i.e. how many excess deaths result from an incremental increase in temperatures that result from an incremental increase in 2020 emissions. Figure 4A shows that temperature-related mortality projections are highly convex in global average temperatures, i.e. mortality increases at an increasing rate in global average temperature.

When global average temperatures exceed 2 °C, the first derivative is quite steep and increasingly so as the world continues to warm. This is what accounts for the significant MCC. In addition, this gives societies a strong incentive to avoid scenarios where global average temperatures are especially damaging. Thus, we find that optimal climate policy in DICE-EMR results in 2.4 °C warming by 2100 compared to 3.5 °C warming in the DICE-2016 optimal climate policy. It is important to note that recent literature has identified other shortcomings in the DICE model including other issues with the climate-economy damage function and the climate module[61,62]. Besides adding the effect of climate change on mortality and subsequent feedbacks, DICE-EMR takes the rest of the DICE model as given without updating other factors. Therefore, this optimal climate policy should not be interpreted as a definitive optimal policy, but as an update to the DICE optimal policy that accounts for the impacts of climate change on human mortality.

From the perspective of policy, the effect of marginal emissions is more important than the aggregate effect that results from all

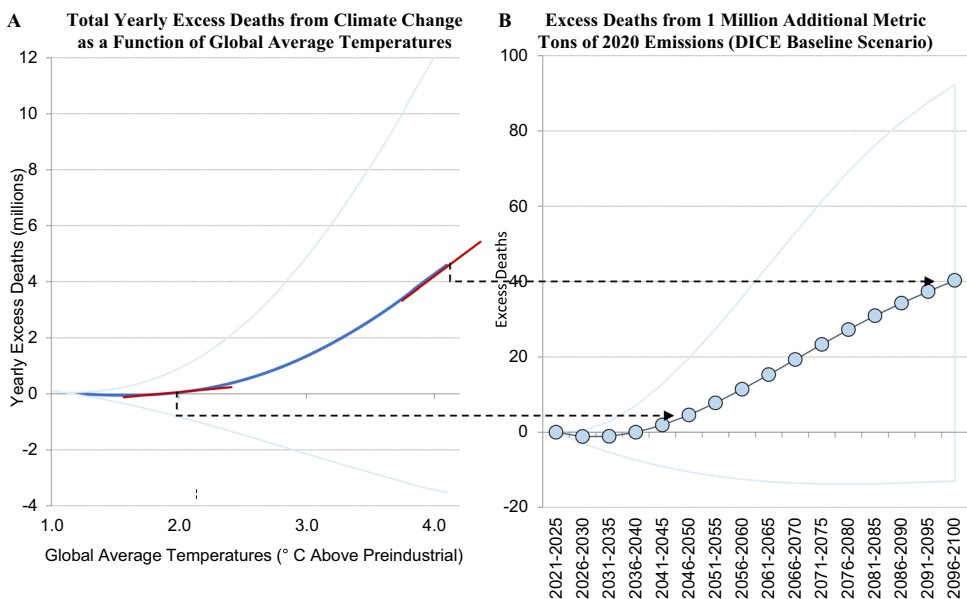

**Fig. 4 The mortality cost of carbon is driven by the convexity of the mortality response.** In both graphs, the high and low lines represent uncertainty with high (>90th percentile) and low (<10th percentile) estimates. **A** Below 2 °C, projected yearly excess deaths from climate change are relatively constant at around 100,000 per year in the central estimate. Above 2 °C, projected yearly excess deaths from climate change increase at an increasing rate in global average temperatures, rising to over four million excess deaths at 4 °C. This implies that the number of excess deaths from a marginal increase in temperatures (the first derivative—represented by the red tangent lines on the graph at 2 and 4 °C) is initially relatively modest but increases substantially with increasing temperatures. **B** Because a significant portion of carbon dioxide emissions remain in the atmosphere for centuries after they are emitted, adding one million metric tons of carbon-dioxide-equivalent emissions in 2020 marginally increases global average temperatures through 2100. The magnitude of excess deaths from marginal 2020 emissions shown in panel B is driven by the steepness of the mortality response curve shown in panel A, which becomes progressively steeper with increasing temperatures. In the 5 years between 2046 and 2050 (when global average temperatures are 2.0 °C above preindustrial in the baseline emissions scenario), the mortality response curve is comparatively shallow, and one million marginal metric tons of carbon-dioxide-equivalent emissions in 2020 are projected to cause five excess deaths in this timespan. In the 5 years between 2096 and 2100 (when global average temperatures are 4.0 °C above preindustrial in the baseline emissions scenario), the mortality response curve is comparatively steep, and these marginal 2020 emissions are projected to cause 40 excess deaths in this timespan. In total, one million additional metric tons of carbon-dioxide-equivalent emissions in 2020 are projected to cause 226 excess deaths in the 80 years between 2020 and 2100 in the baseline emissions scenario. These are concentrated at the end of the century when global average temperatures are highest and marginal changes to temperatures are most damaging.

global economic activity in aggregate across time[63–66]. This study presented and quantified two measures of the effect of marginal emissions: (1) the MCC, which is the effect of marginal emission on excess deaths, and (2) the SCC, which is the full monetized damages from marginal emissions. While DICE-EMR estimates marginal effects in the form of the SCC and the MCC, it can also be used to assess the affects of policy changes that are non-marginal, such as the number of lives saved if countries pursue different emissions targets.

Separate from policy, the MCC and SCC can be useful in informing the decision-making of individuals, households, companies, charities, and other organizations in determining the social impact of the emissions generated by their activities. The emissions contributions of these groups are usually marginal relative to the aggregate emissions of the world economy from the industrial revolution through the twenty-first century. Therefore, the social impact of changes in their activities that either reduce or increase emissions should be quantified using estimates of marginal impacts: i.e. the SCC and the MCC. Because the MCC in the DICE Baseline scenario is $2.26 \times 10^{-4}$ excess deaths per metric ton of 2020 emissions, this implies that if an organization reduces its 2020 carbon dioxide emissions by one million metric tons (roughly equal to the average annual emissions of 35 commercial airliners, 216,000 passenger vehicles, 115,000 homes, and 0.26 coal-fired powerplants in the United States[57,58]), this will save 226 lives in expectation over the course of the twenty-first century. In addition, because the SCC is $258 per metric ton of 2020 emissions, this implies that reducing that same one million

metric tons reduces climate damages by $258 million in monetized net present value terms. Both the SCC and the MCC can be useful for individuals and groups seeking to estimate the social impact of their choices that affect emissions, such as choices around transportation, energy generation, diet, and energy efficiency.

While the SCC is a crucial figure for climate policy, it requires all climate damages to be valued and discounted. To do this, modelers must make subjective ethical choices around how to value non-market damages and how to discount the welfare of future generations relative to current generations. Differences of opinion over how to address these issues can result in substantially different estimates for the SCC, even when the projections of the climatic and socioeconomic consequences of climate change are similar[67,68]. A recent study has suggested that these differences in opinion are so intractable that the SCC has limited value in informing carbon prices[69]. With current techniques, the importance of these ethical choices in driving the results is often obscured because the SCC represents the net effect of all climatic and socioeconomic projections in addition to ethical assumptions. For this reason, we suggest that the best practice should be to provide estimates of the non-market effects of emissions in original units without monetization or discounting in addition to the SCC. This best practice provides greater transparency into the results and empowers users to make their own assumptions on how to value and discount the non-market effects of climate change. In the case of mortality, this estimate is the MCC.

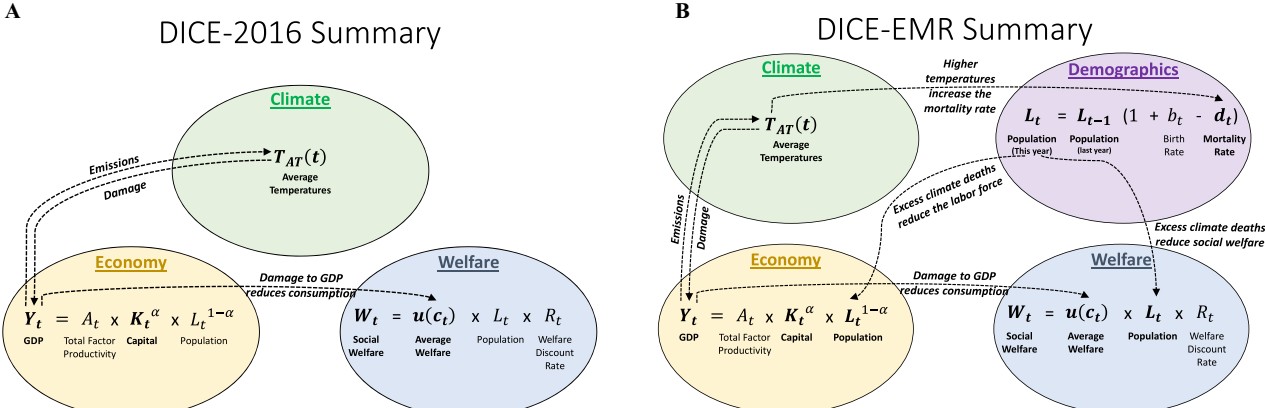

**Fig. 5 A summary of the DICE-2016 and DICE-EMR integrated assessment models (IAMs).** Endogenous components (determined by the model) are in bold. Exogenous components (inputs to the model) are not bold. **A** is a summary of the DICE-2016 model. In this model, the economy affects the climate through emissions and climate only affects society through the damage function that reduces GDP. Our review of[8]—the review study that constructs the DICE-2016 damage function—concludes that mortality costs account for <5% of the damages in the damage function. **B** is a summary of the DICE-EMR model. DICE-EMR takes the rest of DICE-2016 as given and adds a fourth system: demographics. The climate affects the mortality rate through the mortality damage function, which is estimated by a systematic research synthesis of the scholarly literature. This directly reduces welfare due to the welfare cost of these excess deaths, and this effect is calibrated to estimates for value of a statistical life year.

It is important to note that this study has several important limitations. First, there remains significant uncertainty around climate-mortality projections, and the primary specification results are based on central estimates in these projections. Second, the mortality damage function only represents temperature-related mortality; it leaves out potentially important climate-mortality pathways such as the effect of climate change on infectious disease, civil and interstate war, food supply, and flooding due to the limited availability of projections for these pathways in the scholarly literature that sufficiently meet our idealized criteria. Third, this does not consider likely mortality co-benefits of stricter climate policies such as decreases in particulate matter pollution. In future work, DICE-EMR could be combined with IAMs that quantify the mortality co-benefits of stricter climate policy[70] to fully quantify the net effect of climate policy on mortality. Fourth, although the mortality damage function accounts for the effects of defensive adaptation in reducing the impact of climate change on mortality, these adaptations are likely to be costly and DICE-EMR does not directly model the costs of these adaptations. The costs of some adaptations, such as increased spending on air conditioning, are in principle included in the original DICE-2016 climate-economy damage function, although our review concludes that the costs are likely understated (see Supplementary Materials). While future research that reduces the uncertainty around temperature-related mortality projections may increase or decrease our MCC and SCC results, the second and third limitations likely contribute towards understating the MCC and SCC and the fourth limitation likely contributes towards understating the SCC (since it would only effect the SCC and not the MCC). If these limitations were accounted for, they would likely increase the central estimates of the SCC further and result in a more stringent optimal climate policy.

## Methods

**Integrating mortality into integrated assessment.** A high-level summary of the DICE-2016 model is shown in Fig. 5A. This figure shows that DICE-2016 has three major systems: economic, welfare, and climate. It is a global model as it models gross world product and it calculates global average temperatures. Without the climate system, the DICE model is essentially the standard Ramsey–Cass–Koopmans Neoclassical Macroeconomic Model of long-run economic growth[71,72]. William Nordhaus's innovation in creating the original DICE model was integrating macroeconomic and climate models into a single model by

modeling the economy's production of greenhouse gas emissions, the effect of these emissions on global average temperatures, and feedback of higher temperatures back on the economy through the climate-economy damage function. DICE-2016 is useful in informing climate policy by determining the SCC and an optimal path of emissions that maximizes the net present value of social welfare.

To determine population gross of the climate-mortality effect in DICE-EMR, we use data from the 2019 UN World Population Prospects, which projects mortality and fertility rates from 2020 to 2095 (ref. [73]). Population before the climate-mortality effect accumulates according to the following difference equation:

$$L_{t+1} = L_t + b_t L_t - d_t L_t = L_t(1 + b_t - d_t) \qquad (1)$$

where $L_t$ is the population in period $t$, $b_t$ is the fertility rate, and $d_t$ is the mortality rate. Before accounting for the climate-mortality effect, $b_t$ and $d_t$ are determined by the figures given in the 2019 UN report, which makes projections largely based on past trends that do not factor in the likely future mortality effects of climate change (see Supplementary Materials for more details on the UN methodology and projections).

We then incorporate the mortality damage function estimated from the systematic research synthesis described below, $\delta(T_t)$, so that population is now calculated net of climate impacts according to the following difference equation:

$$L_{t+1}(T_t) = L_t(T_t)\{1 + b_t - d_t[1 + \delta(T_t)]\} \qquad (2)$$

Now, the global human population level, $L_t$, is a function of global average temperature, $T_t$, through its effect on the mortality damage function.

**Mortality impacts in DICE-2016.** In DICE-2016, climate change affects society through only one equation: the climate-economy damage function. The climate-economy damage function is a reduced-form equation that represents the portion of economic output lost due to climate change as a function of global average temperatures. Although the DICE-2016 climate-economy damage function is meant to capture all of the market and non-market damages from climate change, in actuality it captures only the damages that are included in the studies used to determine it. It is estimated by fitting a quadratic curve using a median-weighted regression of climate damage projections made by 26 studies chosen from the economics literature[8]. However, most of these 26 studies were heavily de-weighted because they were either superseded by later studies that were also included or they were determined to have poor methods. It is not readily apparent which damages are included or not included without reading the studies individually, an exercise we do for the most heavily weighted studies in the Supplementary Materials. We found that there was significant heterogeneity in the inclusion of mortality impacts: some of the studies used to determine the climate-economy damage function include the impacts of climate-induced mortality while some do not. The studies that do include mortality do so only to a limited extent. Among the most heavily weighted studies, the study that ascribes the highest damages to mortality projects that mortality accounts for 10% of total damages. In addition, this study was done in 1992 and it projects damages only to the United States even though it is used as one of the most heavily weighted studies in estimating the global climate-economy damage function[74]. After reviewing the studies that were used to make the DICE-2016 climate-economy damage function, we conclude that less than 5% of the damages in DICE-2016 come from mortality (see Supplementary Materials for details).

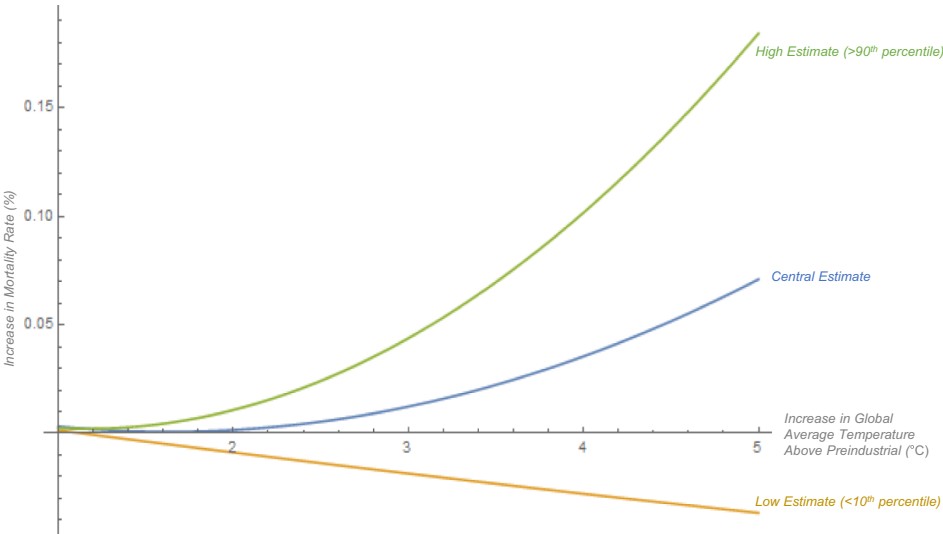

**Fig. 6** Mortality damage function derived from systematic research synthesis estimates the mortality damage function: $\delta(T_t) = \beta_1 T_t + \beta_2 T_t^2$ where $T_t$ is the increase in global average atmospheric temperatures above preindustrial, $\beta_i \in \{\beta_1, \beta_2\}$ are the estimated coefficients, and $\delta(T_t)$ is the % increase in the mortality rate (where for instance 0.05 represents a 5% increase in the mortality rate). See Supplementary Materials for detailed explanation of methods.

**Calculating the MCC in DICE-EMR.** The MCC assesses the marginal mortality effect of carbon emissions in units of excess deaths. It represents the number of excess deaths over some time period from one ton of additional carbon-dioxide-equivalent emissions. It is estimated according to the following equation (see Supplementary Materials for derivation):

$$\text{MCC}(2020) = \sum_{t=2020}^{t=2100} \frac{\partial \delta(T_t)}{\partial T_t} \frac{\partial T_t}{\partial E_{2020}} L_t d_t \qquad (3)$$

This expression is useful for intuition. It shows that the MCC is driven by two factors:

(1) $\partial \delta(T_t)/\partial T_t$: The marginal effect of slightly higher global average temperatures on the mortality, i.e. the first derivative of the mortality damage function $\delta(T_t)$.

(2) $\partial T_t / \partial E_{2020}$: The marginal effect of 2020 emissions on global average temperatures, which is determined by the climate model.

Factor (1) shows why the MCC is sensitive to the convexity of the mortality damage function. $\partial \delta(T_t)/\partial T_t$ is relatively small under the lower temperatures in the first half of the twenty-first century, but because the mortality damage function is highly convex, as the century progresses and temperatures rise past 2 °C, $\partial \delta(T_t)/\partial T_t$ becomes much larger, as shown in Fig. 6. This implies that a marginal emission in 2020 causes significant damage, mostly coming towards the end of the century when temperature levels are higher. This explains why the marginal effect of carbon emissions on excess deaths is surprisingly large compared to what may be expected from the total effect of carbon emissions, shown in Fig. 4A.

The reciprocal of the MCC is equivalent to the metric tons of additional emissions that cause one excess death from 2020 to 2100 on the margin. An MCC of $2.26 \times 10^{-4}$ implies that emitting an additional 4,434 additional metric tons of carbon dioxide in 2020—equivalent to the lifetime emissions of 3.5 average Americans, 146.2 Nigerians, and 12.8 average world people—causes one excess death. Average lifetime emissions are calculated as the 2017 carbon dioxide emissions production per capita[75] multiplied by 2017 life expectancy at birth[76].

**Calculating the SCC in DICE-EMR.** The 2020 SCC is determined by the following equation[1]:

$$\text{SCC}(2020) = \frac{\frac{\partial W}{\partial E(2020)}}{\frac{\partial W}{\partial C(2020)}} \qquad (4)$$

See Fig. 5 for variable names and explanations. $\frac{\partial W}{\partial E(2020)}$ represents the welfare damage from marginal emissions and dividing it by the term $\frac{\partial W}{\partial C(2020)}$ turns this welfare loss into 2020 consumption-equivalent units. Focusing on the damage term (the SCC numerator), the welfare loss in DICE-2016 simplifies to the following equation (see Supplementary Materials for full derivation):

$$\sum_{t=2020}^{t=2510} \frac{\partial u(c_t)}{\partial c_t} \frac{\partial c_t}{\partial E(2020)} L_t R_t \qquad (5)$$

As the equation shows, emissions in DICE-2016 cause damages only through their effect on reduced consumption. $\frac{\partial c_t}{\partial E(2020)}$ is the loss in the average person's consumption multiplied by the marginal utility of consumption, $\frac{\partial u(c_t)}{\partial c_t}$, and then

scaled by the exogenously determined population. The marginal welfare loss from a marginal 2020 emission is determined in each period of the model and then aggregated across time and discounted by the exogenous rate of social time preference, $R_t$.

In DICE-EMR, there is endogenous mortality, and therefore the population term $L_t$ is now endogenous. The damage term in DICE-EMR becomes (see Supplementary Materials for full derivation):

$$\frac{\partial W}{\partial E(2020)} = \sum_{t=2020}^{t=2510} \frac{\partial u(c_t)}{\partial c_t} \frac{\partial c_t}{\partial E(2020)} L_t R_t + \sum_{t=2020}^{t=2510} \frac{\partial L_t}{\partial E(2020)} u(c_t) R_t \qquad (6)$$

This equation can be broken into two terms that are useful for intuition:

(1) *The consumption effect:*

$$\sum_{t=2020}^{t=2510} \frac{\partial u(c_t)}{\partial c_t} \frac{\partial c_t}{\partial E(2020)} L_t R_t \qquad (7)$$

(1) *The welfare effect of mortality:*

$$\sum_{t=2020}^{t=2510} \frac{\partial L_t}{\partial E(2020)} u(c_t) R_t \qquad (8)$$

As in DICE-2016, an additional ton of emissions in 2020 affects social welfare through its effect on consumption as captured in *(1) the consumption effect term*. However, DICE-EMR has an additional *(2) welfare effect of mortality term* that captures the direct loss in welfare resulting from excess deaths caused by climate change. To accurately capture this effect, it is necessary to calibrate the utility function to a value of a statistical life year (VSLY).

We leverage recent methodology[77,78] to calibrate the welfare loss from higher mortality in general equilibrium to VSLY as a multiple of consumption (see Supplementary Materials for details). DICE-EMR is a single representative agent global macroeconomic model, so this is calibrated as a multiple of global average consumption, which is just under $12,000 in 2020.

The structure of DICE-EMR as a single representative agent model has an important implication for valuing loss of life in the SCC: it gives equal weight to deaths no matter where they occur in the world. All lives are valued at the global average level. Alternative methodologies give greater weight to richer individuals who compared to poorer individuals die based on their willingness to pay to avoid a higher probability of death. Because richer individuals have more financial resources, they have a higher willingness to pay to avoid a higher probability of death[79]. The implication of these alternative methodologies is that lives in richer countries (e.g. in Western Europe, North America) are weighed more than lives in poorer countries (e.g. in Africa, South Asia).

For instance, the 2019 Climate Impact Lab paper, which is one of the studies that we use to construct our mortality damage function, monetizes the excess deaths caused by marginal emissions to find a mortality partial SCC that only includes the monetized mortality damages from climate change. In their primary specification, they find that the 2020 mortality partial SCC is $38.1 per metric ton assuming an RCP 8.5 emissions scenario. The major reason for this discrepancy between their results and the SCC results in this study are: (1) they value lost life

years instead of valuing all lost lives at the same value and (2) they apply a different VSLY in each of their different regions that varies with the region's income. In tables H2 and H3 of their paper, they run sensitivities where they show how the 2020 mortality partial SCC varies when they make different assumptions around mortality valuation. When they value all lost lives at the same value and when they value lost lives at a single average global VSLY, as we do, they end up with a mortality partial SCC of $149 (shown in the middle left-hand side of their table H3), which is more in line with our results.

The IPCC states that the approach taken by DICE-EMR—valuing all lives at the same level—is nearer the truth than the approach of valuing the lives of the rich more than the lives of the poor[63]. This approach is also consistent with policies undertaken by national governments: although there are often significant regional heterogeneities in incomes within countries, no national governments currently assign higher values to the statistical lives of richer citizens or lower values to the statistical lives of poorer citizens in cost–benefit analyses. Since our level of analysis is global, we also take this approach.

As discussed in Fleurbaey et al. (2019)[80], another alternative approach that can be used in multi-region multi-agent models is to weigh regions at their local VSLY, and then to apply inequality weights so that poorer regions are given greater priority and are not discounted in welfare terms. Such an approach cannot be taken in the single-region single representative agent setting of DICE. However, as they discuss, using the average VSLY, as we do here, also alleviates the repugnance of discounting the lives of the poor, although in a less rigorous way relative to inequality weights. Future work that uses an integrated assessment model with more regions could take the alternative approach of inequality weights.

DICE-EMR takes an *opportunity cost of life* methodology to valuing the lost life from climate change (see Supplementary Materials section for more details). Lives that are not lived due to climate change whether because they died or because they were not brought into existence in the first place—what philosopher John Broome calls *absences*, see ref. [81] chapters 9 and 10—are counted as a welfare loss in the social welfare function and accounted for in the SCC. Higher mortality leads to lower total social welfare from the opportunity cost of those who could have been alive to enjoy their utility if they or their ancestors had not died as a result of climate-induced mortality. Functionally, however, the magnitude of absences is small relative to deaths in the near-term centuries that are given the most weight in the social welfare function due to discounting (see Supplementary Materials for details).

**A systematic research synthesis of the climate mortality literature**. We conducted a systematic research synthesis of the scholarly literature on the mortality effects of climate change. We established a set of idealized criteria, and our goal was to find studies that met these criteria as well as possible:

1. Provides a projection of the number of excess deaths or the increase in the mortality rate for a specific warming scenario or scenarios.
2. As comprehensive as possible of the human mortality impacts caused by climate change.
3. Mortality estimates are projected net of defensive adaptation.
4. Mortality estimates are aggregated at the global level, or global estimates can be derived from the provided estimates.
5. Published in the last 20 years.

We surveyed 100 candidate studies to determine if they adequately met the criteria described above. Our synthesis met the qualifications for a quantitative systematic research synthesis as specified by Nordhaus and Moffat[8], although we did not follow a standardized protocol such as the PRISMA 2020 statement. See the Supplementary Materials for more detailed discussion of research synthesis. A wide variety of scientific disciplines assess the effect of climate change on human mortality, especially public health, economics, epidemiology, and medicine. To assess the latest scientific understanding of the climate-mortality relationship, we considered papers from all scientific disciplines. More detailed information on the approach and methods is given in the Supplementary Materials.

We found that our idealized criteria for inclusion in the mortality damage function are quite demanding. A common limiting factor was that studies only covered a limited geographic area, as making a full global climate-mortality projection requires a large and comprehensive dataset of human mortality statistics to understand underlying climate-mortality mechanisms (see Supplementary Fig. 1 and Supplementary Table 1 for full details). In addition, of the 100 studies surveyed, few attempted to account for adaptation; all the studies that did were published in 2011 or later. The majority of the studies that account for adaptation project it to have a large role in limiting the damage done from climate change[18,26,41,82].

Although no study perfectly met all five criteria, a few studies came sufficiently close, and these studies were used to construct the mortality damage function. The studies ultimately chosen were a 2014 WHO Report "Quantitative risk assessment of the effects of climate change on selected causes of death, 2030s and 2050s"[26], a 2019 Climate Impact Lab report "Valuing the Global Mortality Consequences of Climate Change Accounting for Adaptation Costs and Benefits"[82], and a 2017 Lancet Planetary Health article "Projections of temperature-related excess mortality under climate change scenarios"[25]. Due to their scope, each of these studies were large multi-institution research collaborations between 16, 17, and 45 authors, respectively.

Many of the 97 studies that were not chosen were used indirectly because their data, methodologies, and results were utilized in these three studies. Each of the three chosen studies featured authors who had worked extensively on the climate-mortality relationship and who had authored some of the other 97 papers that were among those surveyed in the systematic research synthesis. Many studies came sufficiently close to meeting the criteria but were excluded because they were either reused by one of the three studies above (Hales et al. (2014)[26] in particular was largely an agglomeration of past studies) or the methods they developed were later applied to a larger dataset that could more accurately capture the global mortality effects of climate change. A more thorough description of each of these three studies, including their advantages and drawbacks, is provided in the Supplementary Materials.

Even though we used projections from three studies to construct the mortality damage function, conducting the systematic research synthesis allowed us to reach a few conclusions about the literature broadly. After surveying these 100 studies, we found that the consensus was that climate change is likely to increase future mortality rates through a number of channels including the direct effects of ambient heat[16-56], interactions between higher temperatures and surface ozone formation[24,43,46,52,55,56,83-92], changes in disease patterns[16,26,43,45,46,49,50,52-55,93-95], flooding[16,26,43,50,52-56,93-95], and the effect on food supply[16,26,43,45,52,54-56,87,95-98].

We deemed that the three studies that were chosen came sufficiently close to meeting our idealized criteria, but they still have some limitations (see Supplementary Materials for full details). In particular, one of the limitations of the 2019 Climate Impact Lab study and the 2017 Lancet Planetary Health study was that they did not account for longer-term non-temperature-related pathways such as deaths from undernutrition, dengue, malaria, and diarrheal disease. The 2014 WHO study, however, did make projections for these pathways. While our original goal was to present as full of a projection of the mortality risk as possible (idealized criteria #2), we ultimately chose to limit our analysis to temperature-related mortality so that the source of climate-induced mortality in the mortality damage function is clearer. Thus, we used only the portion of the WHO projections from temperature-related mortality and, therefore, our mortality damage function only projects temperature-related excess mortality.

The 2017 Lancet Planetary Health study has a few limitations: (1) It does not provide a full global estimate but instead provides estimates for nine different regions that represent ~40% of the world's population and are mostly in higher-income areas, and (2) It did not attempt to account for adaptation. We addressed (1) by using the methodology described in the Supplementary Materials to convert the 2017 Lancet Planetary Health regional estimates into a global estimate. With respect to (2), the economics literature on climate-mortality adaptation has suggested that in the United States, there has already been significant adaptation to climate change that has ameliorated the mortality effect of hot days, in particular through the adoption of air conditioning[99]. This adaptation has also likely already occurred in other rich regions that have widely adopted air conditioning, such as in Europe, much of the Americas, and some countries in East Asia. Much of the expected future benefit of climate-mortality adaptations can be expected to come from emerging countries[82]. The exclusion of the most vulnerable regions contributes towards understating the future global mortality projection while the exclusion of adaptation contributes towards overstating the future global mortality projection. Despite these limitations, we still decided to use the 2017 Lancet Planetary Health because it appears to be the most global and sophisticated study of the effect of temperature-related excess mortality in the epidemiology literature. A limitation of the 2014 WHO study is that it only projects heat-related mortality. The authors of the study state that they make this choice because the most recent IPCC report concludes that the impacts on health of more frequent heat extremes greatly outweigh the benefits of fewer cold days.

With these limitations in mind, we ran DICE-EMR with (1) an alternative specification in which the mortality damage function does not include the 2017 Lancet Planetary Health study, and (2) an alternative specification in which the mortality damage function does not include the 2014 WHO study. The results from these alternative specifications are shown in Supplementary Figs. 4 and 5. In both of these specifications, the central estimate SCC and MCC were slightly higher. The SCC went from $258 in our primary specification to $295 in alternative specification (1) and to $264 in alternative specification (2) while the MCC went from $2.26 \times 10^{-4}$ in our primary specification to $2.71 \times 10^{-4}$ in alternative specification (1) and $2.38 \times 10^{-4}$ in alternative specification (2).

We construct the mortality damage function by fitting a curve through the central estimates of the projections of these three studies using a weighted regression where each study was given proportional (1/3) weight, and each data point within a study is given proportional weight (described in detail in the Supplementary Materials). We did the curve fitting exercise for six different functional forms including linear and five different non-linear functional forms. While the linear regression yielded a poor fit, all of the non-linear functional forms yielded similar curves that fit the data well (see Supplementary Table 3 and Supplementary Fig. 3). To maintain consistency with the functional form of the climate-economy damage function in the original DICE model, we chose to use a quadratic functional form for the mortality damage function. Mortality projections in warmer scenarios (>3 °C) in these studies were especially damaging, and this is reflected in the mortality damage function (see Fig. 6). In its central estimate, the mortality damage function projects that a scenario in which global average temperatures increase by 4.1 °C causes the mortality rate to increase by 3.8%.

As each of the three studies used to construct the mortality damage function show (2014 WHO, 2019 Climate Impact Lab, and 2017 Lancet Planetary Health), there are expected to be significant heterogeneities in the mortality effect of increasing temperatures in different locations. In general, places that are currently hotter are expected to tend to have more excess deaths, and some places that are currently colder are expected to have net mortality benefits. The studies project that excess deaths from climate change in hotter areas are expected to outweigh the fewer deaths in colder areas, and the net global effect is expected to be an increase in excess global temperature-related mortality. DICE-EMR uses the global projections given by these studies to construct the mortality damage function.

The increase in the global mortality rate projected by these studies is convex in global average temperatures (see Supplementary Figs. 2 and 3). One mechanism for this frequently mentioned in the literature is that extreme hot days make it difficult for humans to thermoregulate themselves: when the wet-bulb ambient temperature exceeds skin temperature (~35 °C), humans can no longer dissipate heat into the environment, causing hyperthermia and greater mortality risk[20,100,101], and extreme hot days have an especially damaging non-linear effect on human mortality. The frequency of extreme hot days is expected to increase exponentially in global average temperatures[102,103]. Places with already hotter climates are projected to be harmed more due to the exponentially greater frequency of extreme hot days and places with colder climates are likely to see some mortality benefits from climate change due to the lower frequency of extreme cold days.

**Uncertainty**. Each of the studies also convey the uncertainty in their projections by providing high and low estimates in addition to central estimates. The Climate Impact lab report provides a high 90th percentile estimate and a low 10th per-centile estimate. The 2017 Lancet Planetary Health report provides a high 97.5th percentile estimate and a low 2.5th percentile estimate. The 2014 WHO report, however, does not provide its high and low estimates statistically, but instead gives estimates as the "highest" and "lowest" estimates. Because of the approach taken by the WHO report, we cannot calculate precision-weighted confidence intervals as is common in other metaanalyses, e.g. ref. [104]. We, therefore, communicate uncertainty in the mortality damage function as "high (>90th percentile)" and "low (<10th percentile)." Like the central mortality estimate, we also produce projections for the high and low estimates through a quadratic weighted regression with the high and low estimates given by the three studies (see Supplementary Materials for more details). We present sensitivities in our MCC and SCC results using these high and low projections. Note that the original DICE-2016 climate-economy damage function does not include uncertainty and only projects a central estimate of climate damages[8]. A recent paper by Gillingham, Nordhaus, and coauthors[105] emphasizes the importance of including uncertainty in integrated assessment. The methodology used to create the mortality damage function represents an improvement over the methodology used to create the original DICE-2016 climate-economy damage function by: (1) including uncertainty and (2) providing documentation for each of the studies that were considered in the systematic research synthesis and providing a reason and coding for why it was used or not used (shown in Supplementary Table 1). In the DICE-2016 systematic research synthesis, some studies that in theory should have met the criteria for relevance, for instance, Burke et al.[106], were excluded without providing a reason for exclusion.

**Estimates of the effect of climate change on fertility**. DICE-EMR incorporates endogenous mortality but not an endogenous fertility; the fertility rate remains exogenously determined by the 2019 UN World Population Prospects. Although climate change is likely to affect the fertility rate[107], the emerging literature on the topic suggests that climate will affect fertility through several different channels, some of which will tend to increase the fertility rate[107] and some of which will tend to decrease the fertility rate[108]. The overall effect of climate on the fertility rate is not yet clear from the literature, even directionally. In keeping with the rest of the analysis, we only model the effect of climate on demographics where the central estimates of the empirical literature are clear directionally. Because we do not explicitly model the effect of climate change on fertility, DICE-EMR should not be viewed as a projection of the effect of climate change on population levels. Instead, it should be viewed as a projection of the effect of climate change on human mortality and the welfare consequences of this effect.

**Reporting summary**. Further information on research design is available in the Nature Research Reporting Summary linked to this article.

## Data availability

The data used to produce the analysis described in this paper and its supplementary materials is available in the following repository: https://www.openicpsr.org/openicpsr/workspace?goToPath=/openicpsr/138881&goToLevel=project under a creative commons 4.0 license.

## Code availability

The code used to produce the analysis described in this paper and its supplementary materials, including the DICE-EMR model and the mortality damage function curve fitting, is available in the following repository: https://www.openicpsr.org/openicpsr/workspace?goToPath=/openicpsr/138881&goToLevel=project under a creative commons 4.0 license. The DICE-EMR model is also available on R. Daniel Bressler's personal website: https://rdanielbressler.com/the-diceemr-model.

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

## Acknowledgements

I thank Leopold Aschenbrenner, Scott Barrett, Tamma Carleton, Floriane Cohen, Marc Fleurbaey, Carolyn Hayak, Geoffrey Heal, Peter Howard, Antony Millner, Duncan Menge, Frances Moore, Jeffrey Sachs, Siddhanth Sharma, Jeffrey Shrader, Rodrigo Soares, Lennart Stern, Charles Taylor, Phillip Trammell, Ana Vicedo-Cabrera, Andrew Wilson, and seminar participants at Columbia University, Oxford University, and UCLA for helpful comments, discussions, and suggestions. Funding from the Earth Institute at Columbia University, the Open Philanthropy Project, the Forethought Foundation, and the Columbia Center for Environmental Economics and Policy is gratefully acknowledged.

## Competing interests

The author declares no competing interests.
