## [Peer Review File · Nature Communications]

Reviewer comments, first round: –

Reviewer #1 (Remarks to the Author):

This paper makes a nice contribution by adding new estimates of mortality impacts of climate change to DICE. This relies on a review of recent empirical literature, although limited to a handful of studies, in spite of a comprehensive review of many more references. The presentation of a direct measure of mortality impact (MCC) is valuable and intuitive.

It would be good if the author could address the following issues:

- Something should be said about the post 2100 impacts. Most IAMs project over 3 to 6 centuries, and it would be useful to know if the impacts for the current century represent most of the impacts or only a small part.
- The scenarios start in 2015 and differ widely in 2020. I am not sure this is very important, but it would be better if the scenarios could start in 2020 and differ afterward only. If this requires too much work of updating DICE, the author should explain why this is not feasible at reasonable cost.
- The VSL numbers are rather high. The author could refer to the work of Hammit and co-authors (such as Robinson, L., Hammitt, J., & O’Keeffe, L. (2019). Valuing Mortality Risk Reductions in Global Benefit-Cost Analysis. *Journal of Benefit-Cost Analysis*). The usual range is rather between 1 and 3 times consumption.
- The methodology about the VSL appears confused. The author is actually talking about the VSLY (life year), not the VSL. This should be corrected and clarified. The above reference can be useful to help the author understand the appropriate methodology.
- The discussion about VSL for rich and poor ignores one possible approach, which has been used and highlighted by authors who incorporate inequalities into such analysis. According to this approach, the best methodology is not to use a single VSL for all deaths, but to use the actual VSL based on willingness to pay, and then give greater priority to the worse off in the population. Depending on the parameters, it may end up giving greater weight to the poor’s longevity (this issue is discussed in Fleurbaey et al. (2019). *The social cost of carbon. The Monist*).

Marc Fleurbaey

Reviewer #2 (Remarks to the Author):

This paper combines a literature review with integrated assessment modeling to investigate the impact of marginal emissions and associated warming on human mortality. It also explores the implications for the social cost of carbon and optimal mitigation. The topic is interesting and the work is thoughtful. I agree with the author that the mortality impacts are likely to be wildly under-represented in current IAMs and that we should therefore expect that the current SCC estimates are too low. However, I have two overarching concerns with the paper. The first is that many of the important analyses are hidden in the Methods or SI, or were not presented. The second is that not enough effort was given to explore the sensitivity of the results to different assumptions. Many of my specific comments will reflect these concerns.

My first set of comments is about the development of the exposure-response function, which forms the basis of the entire project considering that the DICE results are dependent on it. In no particular order, my worries about that aspect of the paper are as follows:

* Some of the main text is a bit misleading about the method. For one thing, it implies that the function is based on 100 studies, when actually only 3 were included. I realize that 100 were reviewed, but a reader could easily think that 100 were actually used in the estimation. In addition, the Methods implies that the three studies more or less fit the inclusion criteria, when actually – as reported late in the SI – they all have some fairly substantial drawbacks. For example, the study by Gasparrini et al. does not account for adaptation. (In fact, that study was used in a recent paper to highlight that that specific paper may NOT be suitable for use in SCC estimates, in part because it doesn't include adaptation – see Scovronick et al. 2019 in *Epidemiology*). I realize that the *Nature Communications* format is unconventional, but these issues are too important to be as concealed as they are.

* What seems to be a key feature driving the dramatic results is the assumption that mortality from temperature exposure increases at an increasing rate (e.g. p4, p17, p18). However, the precise evidence for this important point is not given in detail and largely without appropriate referencing.

* A related issue is the decision to use the quadratic regression. Although damage functions are often assumed to be quadratic, was the quadratic actually a better fit to the "data" than a linear or other potential shape? If not, these other shapes would need to be explored and could change the results substantially. The data points should be on Fig. 4 to allow the reader visually assess the goodness of fit.

To summarize, the paper is dependent on the exposure-response function, yet crucial details of its estimation are either hidden or not provided. This aspect needs much more, and more transparent, treatment. Promoting some of the important details to the main text would help, but more explanation is also needed overall.

The next set of comments is about the results from DICE.

* I had trouble following the modification(s) to the utility function and specifically whether total population size remains a factor in the SCC calculation in DICE-EMR. This is an issue, of course, because combining a total utilitarian utility function with endogenous population means a future death would affect the social cost of carbon through (1) the VSL, (2) labor, and (3) through the SWF merely because there will be less future people. Including all three, it seems to me, may not be appropriate and could imply some double counting. Relatedly, I also worry that an average reader of this paper as is could easily assume that the mortality effect on the SCC is driven entirely through the number of lost lives x the VSL. [It might be helpful to connect this literature to recent papers on the effect of population size on the SCC, such as Budolfson et al. 2019 in *WBER*, Gillingham et al. 2015 in *NBER*, and Scovronick et al. 2017 in *PNAS*. This issue is also pertinent to the question of when people are dying from heat and whether they are beyond reproductive age, because that would not affect the next generation's population size.] If I misunderstood something here, my apologies, but perhaps that indicates that these issues should be more clearly discussed for non-specialists, because they are important.

* The results for the SCC with mortality here are \$265 per ton. How does this compare to the Climate Impact Lab results? My memory is that they also estimate the SCC from mortality, but their findings are rather more modest. If I am indeed remembering correctly, what is the reason for the difference?

* Several studies (e.g. Hansel et al 2020 in *NCC*) have recently reported that updates to DICE 2016 make a big difference to the SCC. At the least it might be worth noting this. While it may not affect the size of the mortality impacts, it will affect the ratio of mortality to non-mortality impacts.

Other miscellaneous comments:

* There are two additional issues that I think should be discussed a bit more carefully. The first is that I do not agree that time preference is irrelevant here. Even though a death is a death whether in 2020 or 2100, decision makers may still have to decide on how to invest in policy, and whether

that (avoided) death occurs sooner versus later is very likely to matter. The second is that the discussion of using VSLs across populations is a bit misleading because inequality aversion could transform different VSLs so they are not different in welfare terms.

* I'm not a huge fan of Figure 1. It gives a newspaper headline-type result, but for the most part it's just a multiplication exercise.

* In Figure 2 there are three scenarios, but in Figure 1, Table 1 and Table 2 there are only two scenarios. This is not a problem per se, but makes for some confusing reading.

* Perhaps I missed it, but what social discount rate is being used in the SCC calculation, and how sensitive are the results to that?

To summarize, I appreciate the author's work and I enjoyed reading this– it is thought provoking. But I would recommend substantial improvements in how the paper is presented, more detail about key assumptions and methods, and more emphasis on sensitivity analyses and teasing apart the factors that are driving these results.

Reviewer #3 (Remarks to the Author):

This is an interesting study aiming at quantifying the cost of marginal CO₂ emissions in terms of future excess mortality. I acknowledge the value and interest of the derived conclusions, in particular, the relevance for policy design. I also agree with the author that the new metric presented here (mortality cost of carbon) overcomes important limitation of the well-known social cost of carbon. Given my expertise, my review will focus on the epidemiological aspects of the study, namely the estimation of the exposure-response and its application for the quantification of the impacts. I apologize in advance if any of my suggestions/comments are not applicable or relevant, as I am not familiar with the DICE models.

I believe that the method applied is an extension of standard methods in climate economics. The main novelty (if I understood well) is the inclusion of the so-called "exposure-response function" used to estimate the excess mortality. The author performed a systematic research synthesis to derive this function from current literature. In my opinion, there are three major concerns: 1) the author derived a UNIQUE exposure-response function, which is not compatible with current epidemiological evidence. First, because as acknowledged by the author, the effect of climate change is due to several factors and pathways, direct and indirect - thus, using a unique function representing the effect of climate change as a whole would not be appropriate. See related comment below. Second, even if the author would have focused on one factor (say direct impacts of temperature on mortality) it has been shown that exposure-response function changes between and even within populations. In conclusion, I believe that the derived impacts would be underestimated in some places, and overestimated in others - which of course, clearly limits the comparability across areas, and thus, the reliability of the derived conclusions.

2) Line 13 page 5: "Central estimates from these three studies were used to run a quadratic weighted regression". This statement is extremely vague. I believe that the estimates would highly depend on the shape of this exposure-response function. Thus, I would encourage the author to extend a bit more how this curve was estimated. Epidemiological studies usually use meta-analytical techniques (see in the same Lancet Planetary Health study used in the manuscript), which can derive summary estimates accounting for uncertainty of the single studies and different sources of heterogeneity.

3) As a side comment, it is not clear from the text whether the author focused on one factor (temperature - as he used the Lancet Plan Health paper which only focuses on temperature), or include other stressors. For example, in line 19 page 12 " Uncertainty is driven by uncertainty in adaptation, uncertainty in the underlying mortality-temperature relationship, and uncertainty in 20 climate model projections." in here, one could assume that the study focuses on the direct mortality impacts attributed to temperature, but there's no additional info before that could clarify

this issue (which is crucial).

Additional comments:

The author mentioned the relevance of adaptation and its implications for mortality projections. As I understood from the text, the way the author accounted for adaptation is through the upper/lower estimates - assuming that (some of) the estimates used to derive these accounted for adaptation. I would first ask the author to clarify this point, and then, if I'm right in my interpretation, this approach is not ideal as we would be mixing very different sources of uncertainty. It would be more appropriate to estimate the contribution of adaptation to these estimates, compare different scenarios of adaptation and discuss on the implications for policy.

Related to the topic above, again if I understood well, the method accounts for demographic changes of the population. As acknowledged in the manuscript, the analysis was restricted by the availability of estimates by age group. The author stated that " Because of this, the MCC captures excess deaths, not lost life years." This is not correct, as even for the computation of excess deaths, age-specific estimates are needed if accounting for demographic changes. Otherwise, the method only accounts for the changes in the total mortality but not its distribution across ages that will change overtime in the future (see Chen et al. 2020 <https://pubmed.ncbi.nlm.nih.gov/32542573/>).

Ana M. Vicedo-Cabrera

Below is R. Daniel Bressler's response to reviewer comments. Reviewer comments are in black, and the response is in red.

Reviewer #1 (Remarks to the Author):

This paper makes a nice contribution by adding new estimates of mortality impacts of climate change to DICE. This relies on a review of recent empirical literature, although limited to a handful of studies, in spite of a comprehensive review of many more references. The presentation of a direct measure of mortality impact (MCC) is valuable and intuitive.

Thank you for your kind words and for your helpful suggestions below that have significantly strengthened the paper.

It would be good if the author could address the following issues:

- Something should be said about the post 2100 impacts. Most IAMs project over 3 to 6 centuries, and it would be useful to know if the impacts for the current century represent most of the impacts or only a small part.

This is an excellent suggestion! I agree that post 2100 impacts are very important and should be discussed. I created an additional section in the supplementary materials called "Post-21st Century Impacts" that discusses this and includes additional figures that show impacts out to 2500. Also, note that, as consistent with the original DICE model, the DICE-EMR SCC calculation includes climate damages through the year 2510 (see equation 3).

- The scenarios start in 2015 and differ widely in 2020. I am not sure this is very important, but it would be better if the scenarios could start in 2020 and differ afterward only. If this requires too much work of updating DICE, the author should explain why this is not feasible at reasonable cost.

Thank you for noticing this and making the suggestion. In the part of the paper where I determine the optimal emissions path that maximizes social welfare (as shown in figure 2), all of the emissions scenarios (including the DICE baseline, DICE-2016 optimized, and DICE-EMR optimized) start from the same 2015 baseline, and then optimal control on the emissions rate starts in 2020.

The reason for this is that in the latest version of the DICE-model (DICE-2016), all scenarios start from the same 2015 baseline, and optimal control of the emissions trajectory also starts in 2020. Therefore, in order to do an apples-to-apples comparison with the optimized DICE-2016 emissions scenario, my optimized scenario must also use the same 2015 baseline and start optimal control of the emissions rate in 2020.

I could attempt to update the DICE-2016 model myself so that all the emissions scenarios start from the same 2020 baseline, and then do optimal control starting in 2025. However, this would essentially amount to me making my own update of the DICE-2016 model to make a DICE-

2020 model. I fear that this would put too much power in my own hands to alter the DICE model in ways that go well beyond extending it to include the effect of climate change on human mortality, and this may undermine the credibility of my results.

The motivation for this paper was to show how just extending the DICE-2016 model to include the effect of climate change on mortality while leaving all other parts of the model the same changes the SCC and the optimal emissions trajectory. If I update DICE-2016 myself, I will have to make a number of assumptions about a wide range of parameter values as they exist in 2020. A number of the key parameters in the DICE-2016 model are the 2015 state values of different economic and climatic variables. To do this, I would have to update these values myself, and since 2020 is not yet finished, I may do a poor job estimating these 2020 values, or I may make estimates that will ultimately be different from the assumptions Nordhaus chooses to make when he comes out with the next version of the DICE model that is updated to make 2020 the baseline year instead of 2015. In this case, it will be hard to disentangle how much of my results are due to different estimations of 2020 parameter values vs. the effect of adding in the effect of climate change on human mortality into the DICE model.

However, your point is well taken, and when Nordhaus comes out with the next version of the DICE model, I will create a version of DICE-EMR that extends that version of the model.

In addition, your question is a good one that other people are likely to have as well. Therefore, I believe that it is important for me to explain my motivation for making this choice, so I added a footnote on page 4 discussing this.

- The VSL numbers are rather high. The author could refer to the work of Hammit and co-authors (such as Robinson, L., Hammitt, J., & O’Keeffe, L. (2019). Valuing Mortality Risk Reductions in Global Benefit-Cost Analysis. *Journal of Benefit-Cost Analysis*). The usual range is rather between 1 and 3 times consumption.

Thank you for pointing this out and for the reference. I added additional description starting on line 12 of page 42 that discusses this reference and discuss the range of variation in VSLY estimates. However, as Robinson et al. 2019 discuss, there really are a wide range of VSLY estimates. In the original version of the paper, I should have provided the specific references that motivate why I use the consumption-multiple figures that I use.

The 2019 Climate Impact Lab study uses a VSLY that is 7.9x the value of per capita consumption in their main specification to calculate the mortality partial social cost of carbon (using the methodology they describe in their appendix H where they convert the US EPA VSL into an implied value per life year, and then apply this value to the rest of the world using unit income elasticity). The Charles I. Jones 2016 *Journal of Political Economy* paper *Life and Growth* (whose methodology I leverage for incorporating endogenous mortality in a general equilibrium setting by calibrating the utility function to VSLY) uses a central VSLY consumption multiple of 3.5. In addition, the 2019 Climate Impact Lab study runs an alternative specification where they use the VSL estimate from the Ashenfelter & Greenstone 2004 *Journal*

of Political Economy study instead of US EPA estimate to calculate the implied value per life year. With this specification, the VSLY is 2.8x yearly per capita consumption.

Also, in Table 2 and throughout the text, I changed the language from “low,” “central,” and “high” VSLY estimates to just 2x, 4x, and 8x yearly per capita consumption, and refer to the 4x as my “main specification.” Since there is such a wide variety of VSLY estimates, I agree that it is important to not claim that 4x is necessarily a central estimate. Table 2 is provided in the introduction so that those who believe that 2x consumption or 8x consumption is a more appropriate VSLY can see how the SCC varies under these assumptions. As this table shows, even if you assume that the VSLY is 2x consumption, the SCC in the baseline emissions scenario is still higher than the original DICE-2016 by a factor of 4.8 (as opposed to a factor of 7.0 if you assume that VSLY is 4x consumption).

I also added into my discussion on page 42 that such a wide discrepancy in the actual value of the VSLY itself is a further motivation for the MCC metric in addition to the other motivations I had already described in the paper.

- The methodology about the VSL appears confused. The author is actually talking about the VSLY (life year), not the VSL. This should be corrected and clarified. The above reference can be useful to help the author understand the appropriate methodology.

Great catch! Yes, this is correct that I am using VSLY and not VSL. Thank you for catching this. I switched the language to VSLY instead of VSL throughout the paper.

- The discussion about VSL for rich and poor ignores one possible approach, which has been used and highlighted by authors who incorporate inequalities into such analysis. According to this approach, the best methodology is not to use a single VSL for all deaths, but to use the actual VSL based on willingness to pay, and then give greater priority to the worse off in the population. Depending on the parameters, it may end up giving greater weight to the poor’s longevity (this issue is discussed in Fleurbaey et al. (2019). The social cost of carbon. The Monist).

This is a very good point. I added additional documentation on this alternative way to deal with heterogenous VSLs in the paper, and mention the paper you cite above in the last paragraph of page 16 and discuss below.

From the perspective of my particular analysis, DICE (and my extension to DICE) deals with a single global representative agent. Such an approach cannot be taken in the single-region single representative agent setting of DICE, but, as you discuss on page 96 of your Monist article, using the average VSL as I do also alleviates the repugnance of discounting the lives of the poor, although in a less rigorous way relative to inequality weights. However, I agree that this is an important point, and I mention that future work that uses an integrated assessment model with more regions could take the approach you mention.

Marc Fleurbaey

Thank you for your thoughtful and helpful review, Marc!

Reviewer #2 (Remarks to the Author):

This paper combines a literature review with integrated assessment modeling to investigate the impact of marginal emissions and associated warming on human mortality. It also explores the implications for the social cost of carbon and optimal mitigation. The topic is interesting and the work is thoughtful. I agree with the author that the mortality impacts are likely to be wildly under-represented in current IAMs and that we should therefore expect that the current SCC estimates are too low.

Thank you for the supportive words, and thank you for your thorough and thoughtful comments and suggestions that have strengthened the paper significantly. I have gone through and addressed all of your comments. It was a good deal of work, but it was worth it because your questions and comments were important to address for readers to be more confident in the results of my paper. You have really helped the paper and I am very grateful.

However, I have two overarching concerns with the paper. The first is that many of the important analyses are hidden in the Methods or SI, or were not presented. The second is that not enough effort was given to explore the sensitivity of the results to different assumptions. Many of my specific comments will reflect these concerns.

Thank you for providing thorough comments below as to how these two major concerns of yours can be addressed. I implement the changes you suggested and address your comments as described below. I hope these changes allay the concerns you have around providing additional information about the analysis and showing that the results are robust to different assumptions.

My first set of comments is about the development of the exposure-response function, which forms the basis of the entire project considering that the DICE results are dependent on it. In no particular order, my worries about that aspect of the paper are as follows:

* Some of the main text is a bit misleading about the method. For one thing, it implies that the function is based on 100 studies, when actually only 3 were included. I realize that 100 were reviewed, but a reader could easily think that 100 were actually used in the estimation.

Thank you for pointing out that this was unclear. I changed language throughout the paper, in particular in the abstract and introduction, to make it clear that I am indeed using three studies to project the mortality damage function (note, for reasons I discuss in response to Reviewer 3

below, I am now calling what was the “mortality response function,” the “mortality damage function.”)

In addition, the Methods implies that the three studies more or less fit the inclusion criteria, when actually – as reported late in the SI – they all have some fairly substantial drawbacks. For example, the study by Gasparrini et al. does not account for adaptation. (In fact, that study was used in a recent paper to highlight that that specific paper may NOT be suitable for use in SCC estimates, in part because it doesn’t include adaptation – see Scovronick et al. 2019 in *Epidemiology*). I realize that the *Nature Communications* format is unconventional, but these issues are too important to be as concealed as they are.

Thank you for pointing out that the discussion of the inclusion criteria and the drawbacks of the studies should have more thorough discussion in the main paper. I agree, and I extended the systematic research synthesis methods section to discuss these issues, in particular on page 13 and also discussed below.

It was a difficult judgment call as to whether it made sense to include the Gasparrini et. al study, in particular because they do not account for adaptation. For the reasons discussed in the paper, I ultimately decided to include Gasparrini in the main specification.

However, you have convinced me that it is important that I run an additional robustness check to show how the results would change when Gasparrini estimates are excluded in the construction of the mortality damage function. I created a new section in the supplementary materials that provides the model results when the Gasparrini estimates are excluded on page 39, and I discuss the results in the main text at the end of the first paragraph on page 13. As you can see, excluding Gasparrini et al. 2017 estimates leads to a slightly higher SCC and MCC. In the DICE baseline emissions scenario, the 2020 SCC increases from \$258 to \$295 per metric ton. The MCC increases from 2.26×10^{-4} per metric ton (implying that 4,434 metric tons of carbon dioxide released in 2020 -- equivalent to the lifetime emissions of 3.5 average Americans -- causes one excess death globally between 2020-2100) to 2.71×10^{-4} lives per metric ton (implying that 3,690 metric tons of carbon dioxide released in 2020 -- equivalent to the lifetime emissions of 2.9 average Americans -- causes one excess death globally between 2020-2100). In addition, the cumulative number of 2020-2100 excess deaths from climate change increases from 83 million to 95 million.

Also, in response to comments from reviewer 3, I decided to simplify the mortality damage function to only include temperature-related mortality. Because of this decision, I am now only using the portion of the WHO estimates that deal with temperature-related mortality. A major drawback of the Gasparrini 2017 and the Climate Impact Lab studies was that they did not account for non-temperature-related mortality. Since I am only now including temperature-related mortality, this drawback no longer holds.

* What seems to be a key feature driving the dramatic results is the assumption that mortality from temperature exposure increases at an increasing rate (e.g. p4, p17, p18). However, the

precise evidence for this important point is not given in detail and largely without appropriate referencing.

Thank you for this helpful comment. I agree that it is important to provide more thorough evidence and referencing for this point. I expanded discussion of this topic in the first paragraph of page 12, and connected the projections of mortality increasing at an increasing rate to the mechanisms that are discussed in the literature.

In addition, I would emphasize that the mortality damage function is essentially just a curve-fitting exercise where I fit a curve through the projections of the mortality increase as a function of the increase in global average temperatures made in the peer-reviewed literature. It is these studies themselves that are projecting that mortality is expected to increase at an increasing rate as a function of global average temperatures (see for instance figure 7 from Carleton et. al 2019 and figure 3 from Gasparini et. al 2017). My study shows the implications of these projections in terms of the effect that marginal emissions have on the SCC and the MCC.

Per your suggestion below, I ran a large amount of additional sensitivity and robustness analysis around the construction of the mortality damage function to show that the large degree of convexity in the mortality damage function is indeed due to the estimates given in the peer-reviewed literature, and not a relic of the decision to use the quadratic functional form. You can see below and in the new figure S3 of my supplementary materials that all 5 nonlinear functional forms in the curve-fitting exercise robustness check yielded a curve quite similar to the quadratic functional form, which is shallow under lower temperatures, but then becomes quite steep under higher temperatures.

* A related issue is the decision to use the quadratic regression. Although damage functions are often assumed to be quadratic, was the quadratic actually a better fit to the “data” than a linear or other potential shape? If not, these other shapes would need to be explored and could change the results substantially. The data points should be on Fig. 4 to allow the reader visually assess the goodness of fit.

Thank you for these good and important questions. When I originally did the analysis, I had run a number of these types of sensitivity checks myself but did not include them in the paper or supplementary materials for purposes of brevity. You are right that I should have shown readers these robustness/sensitivity checks to make my results more convincing, so thank you for raising this point!

As a result, I added an additional section to the supplementary materials (page 36-39) showing these sensitivities across functional forms, and I add discussion of this sensitivity analysis in the second paragraph of page 11. I repeat some of the analysis shown there below for convenience.

A scatterplot of Table S2 showing the central estimate increase in the mortality rate as a function of the increase in global average temperature is shown below.

Fig. S3

We estimate the mortality damage function by fitting a curve through this data. We do this by running a weighted regression where each study is given 1/3 weight, and each data point within a study is given proportional weight. We ran this weighted regression for a number of functional forms, shown in figure S3 and table S3. These include (A) linear, (B) quadratic, (C) 3rd order polynomial, (D) exponential, (E) power, and (F) two-parameter Weibull. As figure S3 and table S3 show, the linear curve (A) produces a relatively poor fit. Each of the non-linear functional forms (B-F) produce similar curves that provide an excellent fit. To maintain consistency with the functional form of the damage function in the original DICE model, we chose to use the quadratic damage function.

Table S3. Mortality Damage Function Functional Form Sensitivities. Each of the non-nonlinear functional forms provide a strong and similar fit.

Fit Type	Equation	R-square	Adjusted R-square
A. Linear Fit	$y = 0.01719x - 0.03087$	0.841	0.826
B. Quadratic Fit	$y = 0.006044x^2 - 0.01878x + 0.01394$	0.970	0.965
C. Third Order Polynomial Fit	$y = 0.0003961x^3 + 0.002631x^2 - 0.01006x + 0.007402$	0.971	0.961
D. Exponential Fit	$y = 0.0004684 \cdot \exp(1.03x)$	0.960	0.956
E. Power Fit	$y = 0.0001811x^{3.745}$	0.968	0.965
F. Two Parameter Weibull Fit	$y = (4.222e-05) \cdot 4.719 \cdot x^{(4.719 - 1)} \cdot \exp(-4.222e-05 \cdot x^{4.719})$	0.968	0.965

Figure S3. Mortality Damage Function Functional Form Sensitivities.

To summarize, the paper is dependent on the exposure-response function, yet crucial details of its estimation are either hidden or not provided. This aspect needs much more, and more transparent, treatment. Promoting some of the important details to the main text would help, but more explanation is also needed overall.

Thank you for raising this general set of concerns and for your specific suggestions above. Thanks to your suggestions, I have now added a good deal of additional analyses with alternative specifications, robustness checks, and more thorough description of the analysis to the paper. I believe that doing these exercises has made the results much more convincing and

credible. Thank you!

The next set of comments is about the results from DICE.

* I had trouble following the modification(s) to the utility function and specifically whether total population size remains a factor in the SCC calculation in DICE-EMR. This is an issue, of course, because combining a total utilitarian utility function with endogenous population means a future death would affect the social cost of carbon through (1) the VSL, (2) labor, and (3) through the SWF merely because there will be less future people. Including all three, it seems to me, may not be appropriate and could imply some double counting. Relatedly, I also worry that an average reader of this paper as is could easily assume that the mortality effect on the SCC is driven entirely through the number of lost lives x the VSL. [It might be helpful to connect this literature to recent papers on the effect of population size on the SCC, such as Budolfson et al. 2019 in WBER, Gillingham et al. 2015 in NBER, and Scovronick et al. 2017 in PNAS. This issue is also pertinent to the question of when people are dying from heat and whether they are beyond reproductive age, because that would not affect the next generation's population size.] If I misunderstood something here, my apologies, but perhaps that indicates that these issues should be more clearly discussed for non-specialists, because they are important.

Thank you for these incisive and thoughtful questions and comments here. I added additional explanation throughout the manuscript on the first paragraph of page 4, the second paragraph on pages 17, and I added an additional section to the supplementary materials that discusses this in similar depth to my discussion below on pages 44-45. In past versions of this paper and in prior presentations, I had more discussion of these points on population size and population ethics, but I ended up cutting most of this in the Nature Communications submission given the need to focus the paper on a more limited number of topics given space constraints. However, your feedback has convinced me that these topics are important enough to be discussed in the main manuscript.

In terms of deriving the utility function that can be used in a general equilibrium setting with endogenous mortality, I added additional explanation to the supplementary materials starting on the bottom of page 41 that I hope makes the methodology clearer. In addition, it is important to emphasize that I am leveraging existing methods that are already widely used in macro/health economics to deal with endogenous mortality in a general equilibrium setting. I am leveraging the methodology developed by Robert Hall and Chad Jones in their 2007 QJE paper that has now been used extensively in Macroeconomic papers but has not been used yet (as far as I've seen) in integrated assessment models. You can see that my derivation of the DICE-EMR Critical Level Isoelastic Utility Function section in the supplementary materials is essentially the same as Chad Jones' recent JPE paper (*Life and Growth* 2016) that also uses this Methodology (described primarily on page 542 and 546 of that paper). Jones' purpose in his 2016 article was to consider that new technologies that improve economic growth sometimes also can have negative mortality impacts (e.g. he mentions pollution, fossil fuels causing climate change, nuclear accidents, and technologies that create conditions that increase transmission of global pandemics), and he explored the tradeoff between growth from new technology vs. the possible

mortality impacts of such technology in a general equilibrium macroeconomic setting. Given the similarity to my goal here, I leveraged his methodology. I added additional explanation to the section where I derive the utility function explaining that I am just using Jones' method verbatim here to make that clearer.

The advantage of this methodology is that it allows you to consider the welfare-cost of higher mortality in a general equilibrium setting with endogenous mortality. The value of life using this methodology – and in other literature in this tradition e.g. Becker, Philipson, and Soares 2005 AER – is the utility that the agent receives while living. As is standard in this literature, the value of death is normalized to 0. Therefore, the level of the utility function matters a great deal, because this determines the value of a year of life relative to a year in which that life does not exist, and thus the utility function is calibrated to VSLY. Equations (S1) – (S3) show how the utility function is calibrated to VSLY, and how this then implies a critical level of consumption above which bringing in a life would be net positive in the social welfare function (see also Jones 2016 542-544). How positive this life lived would be is determined by the level of consumption and the η parameter, which determines the amount of utility that the agent gets when consuming above this level.

This methodology is designed to work with the population term in the SWF so that lives above this critical level that are not lived due to climate change – whether because they died or because they were not brought into existence in the first place (what John Broome calls *absences*; see *Climate Matters* 2012, chapters 9 and 10) are counted as a welfare loss in the SWF. This can be thought of as an *opportunity cost of life* methodology. Higher mortality leads to lower total social welfare from the opportunity cost of those who could have been alive to enjoy their utility if they or their ancestors had not died as a result of climate-induced mortality.

The critical level utilitarian approach taken by DICE-EMR includes both the welfare loss from direct deaths and absences and both of these are valued at the opportunity cost of the life that could have been lived if climate change did not alter the human population through its effect on mortality. The alternative approach you mention of calculating the welfare loss from excess climate deaths based on the VSL accounts for the welfare-loss from death, but not for the welfare-loss from absences. This is not necessarily a bad thing, as there is a good deal of argument and discussion in the population ethics literature about how to treat absences. Derek Parfit in *Reasons and Persons* argues for counting absences, especially in scenarios with significant mortality and John Broome leans towards this approach (see *Climate Matters* p. 89). However, this approach of counting absences goes against an intuition that a lot of people seem to have (what John Broome calls the *Intuition of Neutrality*, see *Climate Matters* p. 83), though Broome argues that the Intuition of Neutrality is false. Also, the inclusion of absences in an SWF is a primary motivation for those that are working on preventing existential risks because of the large value of the future that may be lost (see e.g. Toby Ord *The Precipice* 2020 and Nick Bostrom *Existential Risk Prevention as Global Priority* 2013).

Functionally within the DICE model, the magnitude of absences is small relative to deaths in the near-term centuries that are given the most weight in the DICE SWF due to discounting. In the 21st century in the baseline emissions scenario, the ratio of deaths to absences is 5:1. By the end

of the 22nd century, the ratio of deaths to absences is 2:1. By the end of the 23rd century, absences finally catch up to deaths and the ratio is 1:1. However, because DICE discounts utility at an annual rate of 1.5%, a util at the end of the 23rd century is worth 1.4% the value of a present-period util in the SWF.

You are correct that the age of people dying is important from the perspective of absences because if deaths are more skewed towards people that are not of reproductive age (the old and young), then there would be fewer absences than we would expect if deaths are distributed uniformly. As I discuss on the last paragraph of page 13, I am currently limited by the availability of age-specific mortality projections in the literature that I can use to construct the DICE-EMR mortality damage function. However, more projections of this sort are expected to be produced soon (Antonio Gasparrini told me, for instance, that his team is working on this), so this is something that can be added in future work (also see point in next paragraph)

In terms of how population levels affect the SCC in the DICE-EMR framework, there would be a similar dynamic at play as described in the Scovronick et. al 2017 PNAS article, where the SCC would tend to be higher with larger population because more people would be harmed by the marginal emission, both in terms of people harmed by lost consumption and a larger number of people that would die/be absent due to a marginally higher mortality rate in future years due to the marginally higher level of warming from the marginal emission.

In terms of how the SCC using the DICE-EMR framework would vary relative to the alternative framework you mention of calculating the welfare loss from excess climate deaths based on the VSL, there would be two competing effects: (1) Because the alternative methodology has exogenous population, the SCC would tend to be more damaging in DICE-EMR because the population is larger for the reasons mentioned above, and (2) Because both deaths and absences are accounted for as welfare loss, this would tend to make marginal emissions more damaging for a given population level. Which of these effects ends up being larger depends on a number of factors including the emissions scenario and subsequent level of warming and subsequent number of climate-related deaths, discounting (higher discounting implies absences are less important since they only accumulate at large levels further in the future), VSL and VSLY assumptions, and more. It would be an interesting and useful exercise to compare the two approaches, as they are both reasonable approaches to population ethics. However, given that I already seem to be pushing the limits of content that can fit into an article in the Nature Communications format, perhaps it is better to clearly state my assumptions and discuss these implications clearly in the paper as I have, and to suggest that I will explore this in future work? In particular, if I were to add another method of calculating the SCC into this paper, I would have to duplicate all of the results in my figures and tables for both methods, and then spend a good deal of manuscript space discussing the difference between the methods, and the reasons for this (as the length of my response here already attests to). But, this would be a great next paper to work on, and I mention in the last paragraph of page 45 that this is something I plan to explore in future work.

* The results for the SCC with mortality here are \$265 per ton. How does this compare to the Climate Impact Lab results? My memory is that they also estimate the SCC from mortality, but their findings are rather more modest. If I am indeed remembering correctly, what is the reason for the difference?

In the primary specification (i.e. the one they give in their abstract), Carleton et. al 2019 find that that 2020 mortality partial social cost of carbon is \$38.1 per metric ton assuming an RCP 8.5 emissions scenario. They also run a number of alternative specifications, which are given in their tables H2 and H3. To get to the \$38.1 in their main specification, they made a few important assumptions as it relates to the valuation of mortality damages: (1) they value the lost life years instead of valuing all lost lives at the same value, and (2) each of their 24,378 regions are given different VSLs based on their income differences. In tables H2 and H3, they run sensitivities where they show how the 2020 mortality partial social cost of carbon varies when they make different assumptions around mortality valuation. When they value all lost lives at the same value and when they value lost lives at a single average global VSL, they end up with a mortality partial social cost of carbon of \$149 (shown in the middle left-hand side of their table H3), which is more in line with my results. As I discuss in the paper the last paragraph of page 16 to the top of page 17, because of the structure of the DICE model and the availability of estimates in the literature, DICE-EMR values all lost lives at the same value regardless of age or the income or location of the person dying.

* Several studies (e.g. Hansel et al 2020 in NCC) have recently reported that updates to DICE 2016 make a big difference to the SCC. At the least it might be worth noting this. While it may not affect the size of the mortality impacts, it will affect the ratio of mortality to non-mortality impacts.

This is correct and I actually did make this point in the paper citing Hansel et. al 2020 as well as Howard and Sterner 2017 in the second to last paragraph of my introduction. I think it's hard to see because of the Nature Comms citation style, so I added a bit more information to make it clear that I am referring to these papers.

Other miscellaneous comments:

* There are two additional issues that I think should be discussed a bit more carefully. The first is that I do not agree that time preference is irrelevant here. Even though a death is a death whether in 2020 or 2100, decision makers may still have to decide on how to invest in policy, and whether that (avoided) death occurs sooner versus later is very likely to matter.

Thank you for raising this point. I agree, and I added in the point that decision-makers may find the timing of deaths useful in the first paragraph of page 21, which underscores why the figure 6b chart is useful as a supplement to the MCC number.

The second is that the discussion of using VSLs across populations is a bit misleading because inequality aversion could transform different VSLs so they are not different in welfare terms.

This is another important point that I agree with. I added in discussion of this alternative methodology in the last paragraph of page 16, citing the Fleurbaey et. al 2019 article in the *Monist*.

From the perspective of my particular analysis, DICE (and my extension to DICE) deals with a single global representative agent. Such an approach cannot be taken in the single-region single representative agent setting of DICE, but, as Fleurbaey et. al discuss on page 96 of their *Monist* article, using the average VSL as I do also alleviates the repugnance of discounting the lives of the poor, although in a less rigorous way relative to inequality weights. However, I agree that this is an important point, and I mention that future work that uses an integrated assessment model with more regions could take the approach you mention.

* I'm not a huge fan of Figure 1. It gives a newspaper headline-type result, but for the most part it's just a multiplication exercise.

Yes, I've come to agree that Figure 1 has too much extraneous information. As a result, I cut out a good deal of the charts in the figure and added a new figure that I think is more helpful, which is the new figure 3.

* In Figure 2 there are three scenarios, but in Figure 1, Table 1 and Table 2 there are only two scenarios. This is not a problem per se, but makes for some confusing reading.

Thank you for this piece of feedback. I agree that it is confusing, so I've relabeled things to hopefully make it clearer. In Table 1, Table 2, and Figure 1, I've now more clearly labeled the scenarios to be "DICE-EMR Optimized Emissions Scenario" and "DICE Baseline Emissions scenario" so that the labeling lines up with Figure 2. Before it was unclear if the "Optimized" emissions scenario was referring to "DICE-2016 Optimized" emissions scenario or "DICE-EMR Optimized" emissions scenario, so hopefully now it is more clear.

* Perhaps I missed it, but what social discount rate is being used in the SCC calculation, and how sensitive are the results to that?

I use Nordhaus's DICE-2016 discounting ($\eta = 1.45$ and $\rho = 1.5\%$). Like all IAM results, the SCC is quite sensitive to the discount rate.

I agree that it is important to discuss and show these sensitivities, so I added a new section to the supplementary materials, "Discounting Sensitivities," that does this. The results of this sensitivity analysis are shown in figure S4.

To summarize, I appreciate the author's work and I enjoyed reading this— it is thought provoking. But I would recommend substantial improvements in how the paper is presented, more detail about key assumptions and methods, and more emphasis on sensitivity analyses and

teasing apart the factors that are driving these results.

Thank you for your kind words. I really appreciate the detail of the comments that you provided. It was a good deal of work to address all of your comments, but it was worth it because your questions and comments were important to address for readers to be more confident in the results of my paper. Your suggestions have really helped the paper and I am very grateful.

Reviewer #3 (Remarks to the Author):

This is an interesting study aiming at quantifying the cost of marginal CO₂ emissions in terms of future excess mortality. I acknowledge the value and interest of the derived conclusions, in particular, the relevance for policy design. I also agree with the author that the new metric presented here (mortality cost of carbon) overcomes important limitation of the well-known social cost of carbon.

Thank you for your kind words.

Given my expertise, my review will focus on the epidemiological aspects of the study, namely the estimation of the exposure-response and its application for the quantification of the impacts. I apologize in advance if any of my suggestions/comments are not applicable or relevant, as I am not familiar with the DICE models.

Your comments below have been very helpful, and as I result I have made a number of changes that have significantly strengthened the paper, and I believe have made it more intelligible to research communities outside of those in economics that are not used to working with reduced-form climate-economy integrated assessment models.

I believe that the method applied is an extension of standard methods in climate economics. The main novelty (if I understood well) is the inclusion of the so-called “exposure-response function” used to estimate the excess mortality. The author performed a systematic research synthesis to derived this function from current literature. In my opinion, there are three major concerns:

1) the author derived a UNIQUE exposure-response function, which is not compatible with current epidemiological evidence. First, because as acknowledged by the author, the effect of climate change is due to several factors and pathways, direct and indirect – thus, using a unique function representing the effect of climate change as a whole would not be appropriate. See related comment below. Second, even if the author would have focused on one factor (say direct impacts of temperature on mortality) it has been shown that exposure-response function changes between and even within populations. In conclusion, I believe that the derived impacts would be underestimated in some places, and overestimated in others – which of course, clearly limits the comparability across areas, and thus, the reliability of the derived conclusions.

Thank you for this context on your perspective reviewing this study as an epidemiologist. After reading through all of your comments it became clear to me that I need to make a number of changes to the analysis and text, in particular, so that the work is more clear for research communities that are not economists and that do not work with IAMs like DICE. I summarize these changes below. In the text, these changes are discussed throughout the now expanded systematic research synthesis section on pages 9-14.

1) I changed the mortality damage function so that it only includes temperature-related mortality impacts (I am now referring to what I had been referring to as the “*mortality response function*” as the “*mortality damage function*”... see point 3 below for more details).

Before, I was using the full mortality estimates as given from the 2014 WHO study, the 2019 Climate impact Lab study, and the 2017 Gasparrini et. al study. Of these three studies, the 2014 WHO study is the only one that considers non-temperature-related mortality. As you note below, Gasparrini et. al 2017 only includes temperature-related mortality impacts. As I discussed in the supplementary materials, although the 2019 Climate Impact Lab study does not devote much discussion to explaining the pathways that they capture or do not capture in their reduced-form econometric approach, their approach should just be capturing temperature-related mortality because their econometric strategy exploits historical variations in temperatures to find a relationship between mortality and temperature in regions across the globe. Indeed the co-director of the Climate Impact Lab, Michael Greenstone, confirmed this when he gave testimony on this project to the US Congress a few months ago (video available at https://youtu.be/N8nCZC0_yxU, his explanation of this topic starts at 51:35).

In addition to temperature-related mortality attributable to climate change, the WHO study also accounts for deaths from undernutrition, malaria, dengue, and diarrheal disease. Originally, I had used the full WHO estimate of mortality attributable to climate change, which includes mortality through all five of those pathways. In integrated assessment modeling, this sort of choice is common because the goal of integrated assessment modeling is to produce as full of an estimate as possible from the current peer-reviewed literature on projections of the effects that climate change is expected to have on society to derive a social cost of carbon, even if there is heterogeneity in terms of how fully the studies account for different pathways. This is in fact the approach taken by William Nordhaus when he estimated the original DICE climate-economy damage function. There is significant heterogeneity in the economic sectors and damage sources accounted for in the studies that he used to estimate the original DICE economic damage function (e.g. some studies accounted for damages from sea-level rise, and some did not).

However, I think you raise a good point that my results would be clearer if I chose a specific climate-mortality pathway. The WHO study is the only study of the three that attempts to account for mortality from non-temperature-related pathways. Therefore, I have decided to include only the WHO’s projection of temperature-related mortality and to leave out their estimates from the other pathways so that my mortality damage function only represents temperature-related mortality. The projections from the Climate Impact Lab and Gasparrini et. al 2017 remain unchanged because they represent only temperature-related mortality.

I have now updated and recoded the mortality damage function so that it only includes temperature-related mortality. I reran the model and updated all of my charts and figures. Now, the results only include temperature-related mortality damages. This ultimately does not change the results too much since the WHO study is the only study that attempted to estimate non-temperature-related mortality. However, I think that making the choice to account only for temperature-related mortality makes my results much clearer, particularly to research communities outside of economics that are more focused on pathways and mechanisms.

2). Thank you for raising the point about heterogeneities in the mortality effect of climate change across regions and populations. It is now clear to me that I did a poor job of conveying how I accounted for these heterogeneities in the original version of the paper. I added significant discussion to the text (in particular on page 11), and I discuss this in more detail below.

It is correct that there is significant heterogeneity between regions and within populations in terms of the expected effect of climate change on human mortality. However, as consistent with all reduced form climate-economy integrated assessment models, I rely on the studies themselves to account for these granular heterogeneities in projections of outcomes. What I use as an input to my model are global projections that represent the net global mortality effect (in units of the percentage increase in the mortality rate) accounting for all of these heterogeneities. Because the DICE model is a single region (that single region being the globe) climate-economy macroeconomic model with a single representative agent, it is necessary for the mortality damage function to be at the same level of analysis as the DICE model itself (i.e. the global level), with the proviso that this global estimate accounts for these granular sources of heterogeneity.

The DICE model is not made to compare different regions or areas. It is made to estimate the social cost of carbon and optimal climate policies by comparing the aggregated global costs and benefits associated with different emissions trajectories and emissions decisions. The purpose of the mortality damage function is simply to produce a best estimate of the expected effect of climate change on the global mortality rate in different emissions scenarios based on the projections available in the peer-reviewed literature.

For example, the Climate impact lab report (Carleton et al 2019) divides the world into 24,378 regions. For each of these 24,378 regions, they estimate the mortality impact of climate change in different climate scenarios accounting for heterogeneities in climate, economic development, expected levels of adaptation, and demographic structure. They find that, due to heterogeneities in these various factors, there are significant differences around the world in terms of the projected impact of climate change on mortality. For instance, they find that a much higher projected mortality rate in Accra, Ghana compared to Oslo, Norway, where the mortality rate is expected to decrease due to climate change (see the first full paragraph on page 6 of their paper). They also derive 24,376 additional mortality estimates in the 24,376 other regions. While they make specific mortality projections for all 24,378 regions, they also make a global projection for the increase in the mortality rate in RCP 8.5 and RCP 4.5, which is shown in their figure 7. Their global projection is the net global mortality effect accounting for the various

heterogeneities in all 24,378 regions that they analyze. Because DICE is a single-region global model, it is the output from figure 7 that I use to estimate the mortality damage function.

Similarly, the 2014 WHO report accounts for heterogeneities between regions and it provides regional projections for 21 different regions. They also provide a global estimate, which is the number of projected excess deaths globally in different scenarios, which is just an aggregation of these regional effects. As with the Climate Impact Lab report, I use the global estimates as inputs to the mortality damage function estimation.

While it is straightforward to get input data from the 2019 Climate Impact Lab Report and the 2014 WHO study because I can just directly use the global projections given in their studies, the Gasparrini et. al 2017 report does not give global projections, so, therefore, it requires more analysis and assumptions to estimate a global mortality impact from this study. The Gasparrini et. al article instead gives projections in 9 different global regions. I use the methodology described on pages 33-34 of the supplementary material to estimate the global mortality impact from the Gasparrini study. Note that my methodology that converts the 9 regional Gasparrini estimates to a single global estimate, while cruder than using the WHO and Climate Impact Lab global estimates provided by the studies themselves, still represents a significant improvement over the current state of the art in the most widely used integrated assessment models. For more details on this, see Veronika Huber et al's 2017 article in Climatic Change (Huber, Ibarreta, and Frieler 2017).

As I discuss in the manuscript on page 13, the WHO and the Climate Impact Lab studies readily passed the bar in terms of meeting the criteria for inclusion in the estimate of the mortality damage function. The Gasparrini et. al 2017 study, however, was more borderline, in part because it did not produce global mortality estimates. I ultimately made the judgment call that I would indeed include it in my main specification (where I generate global estimates using the methodology described in the supplementary materials) because it appeared to be the most global and sophisticated study of the effect of temperature-related excess mortality in the epidemiology literature. However, recognizing that the Gasparrini paper had limitations in terms of being usable in this study relative to the WHO and Climate Impact Lab paper, I also added an additional run of the model that excluded Gasparrini estimates from the mortality damage function estimation. The results of this are discussed on the bottom of the first paragraph on page 13 and shown in the supplementary materials on page 39. When the Gasparrini estimates are excluded, the SCC and MCC go up and optimal climate policy becomes more stringent, as you can see in figure S4. For now, I am leaving the WHO + Climate Impact Lab + Gasparrini estimates as the main specification while the WHO + Climate Impact Lab specification is in the supplementary materials. Please let me know if you disagree (as you are an author of that study and would be in a good position to comment), and I am happy to exclude the Gasparrini estimates in the main specification if you think it is appropriate.

3) It is also now clear to me that my original terminology of a “*mortality response function*” is confusing because it sounds similar to the “*exposure-response function*” concept that is commonly used in epidemiology. However, what I had been calling the “*mortality response*

function” is actually analogous to the “*economic damage function*” that was used by William Nordhaus in the original DICE model (see Nordhaus and Moffat 2017) as opposed to the “*exposure response function*” concept that is commonly used in epidemiology. Therefore, I changed the terminology I am using from “*mortality response function*” to “*mortality damage function*.”

It seems that the way you interpreted the original version of my paper is consistent with the way epidemiologists interpret exposure response functions: i.e. the exposure response function represents the change in some outcome as a function of the level of exposure to some environmental factor for a given (sub)population in a particular place. In this case, this would be the increase in the mortality rate as a function of the increase in temperatures in a particular place. This is a natural interpretation given that this is the definition of the term “*exposure response function*” as it is used in epidemiology. With this interpretation, it would imply, for instance, that the exposure response function (in terms of an increase in the mortality rate as a function of the increase in average temperatures) is the same in Ghana as it is in Norway, which as discussed above, we know is not correct.

However, this is not what the mortality damage function represents in DICE-EMR. The mortality damage function is simply a mapping from a specific warming scenario (which is represented by an increase in global average temperatures) to a projected increase in the global mortality rate. This function is estimated by fitting a curve through the projections that have been made in the peer-reviewed literature – see point 2 above for more detail and how this mapping does indeed provide an accurate estimate of the net global mortality effect of climate change accounting for heterogeneities even though it is a single global function.

In general, the purpose of climate-economy integrated assessment modeling is not to produce new information on the relationship between climate and economic output (in the case of the original DICE economic damage function) or climate and mortality (in the case of the mortality damage function in DICE-EMR), but to synthesize existing information to inform policy. IAM damage functions are just a mapping from the increase in global average temperatures to a change in some socioeconomic variable of interest that would affect the welfare of human society. IAM modelers do not make their own original projections, but they collect projections from the peer-reviewed literature, and then fit a curve through these projections, and use this as an input to the model.

In summary, I believe that the mortality damage function may represent something simpler than you had in mind when you read the paper originally (due to my poor original choice of wording and poor explanation!). It is simply a mapping of global projections made in the peer-reviewed literature to the associated increase in global average temperatures in the scenario in which those projections are made. The mortality damage function is no better or no worse than the studies used to construct it (provided that a curve can be well-fitted to the data, which in this case the curve does fit the data very well as discussed in response to your other comment below). If the studies used did a good job of estimating the temperature-mortality relationship, and they accounted for the complexities and heterogeneities in this relationship to get to their global estimates of the increase in the mortality rate in specific climate change scenarios, then the mortality damage function does as well.

I would also emphasize that the results of DICE-EMR, as is true with any other climate-economy integrated assessment model, are not meant to be set in stone. They are meant to be updated regularly so that the latest findings of the peer-reviewed literature are represented in the social cost of carbon estimate used in policy. The goal of IAMs is to provide policymakers with a social cost of carbon that accounts for our latest scientific understanding of the effect that climate change is projected to have on society. Under the Obama administration, the social cost of carbon that was used in regulatory analysis in the United States was determined by an interagency process between the Environmental Protection Agency, Department of Energy, Department of the Interior, and other government agencies that collaborated with outside experts to estimate the social cost of carbon (see https://www.epa.gov/sites/production/files/2016-12/documents/sc_co2_tsd_august_2016.pdf). A review of this process was done by the National Academies of Sciences in their 2017 report (<https://www.nationalacademies.org/our-work/assessing-approaches-to-updating-the-social-cost-of-carbon>), and they emphasized the importance of making sure that the social cost of carbon represents the latest projections of climate impacts on society given the latest scientific literature, and also that there is a process for damage estimates in the SCC calculations to be updated regularly. They also specifically mentioned that the IAMs used to estimate the SCC should be updated immediately to reflect the latest scientific projections of the effect of climate change on mortality (which is a principal reason for this analysis). This interagency working group was stopped by the Trump administration (NYU's Institute for Policy Integrity has a good overview of this history/process available here: <https://costofcarbon.org/faq/what-is-the-scc>), but it is expected to start up again in 2021 under the Biden administration. The interagency working group used the DICE model, along with two other models that have similar damage functions (PAGE and FUND) to project the SCC. As discussed in the paper, these studies are largely leaving out the effect of climate change on mortality in calculating the social cost of carbon, and are therefore significantly underestimating it. Thus, this study will likely be used to update the SCC in the interagency working group in the Biden administration along with other studies that have extended these three IAMs to account for our latest understanding of climate impacts (e.g. <https://www.nature.com/articles/s41467-017-01792-x> <https://www.nature.com/articles/s41467-019-09499-x> <https://www.nature.com/articles/s41558-020-0833-x>). Without this study, the effect of climate change on human mortality may continue to largely be left out of SCC estimates. The Biden interagency working group process will likely update SCC estimates frequently, and the findings from this study will continue to be updated as new research continues to be done on the mortality projections of climate change.

2) Line 13 page 5: “Central estimates from these three studies were used to run a quadratic weighted regression”. This statement is extremely vague. I believe that the estimates would highly depend on the shape of this exposure-response function. Thus, I would encourage the author to extend a bit more how this curve was estimated. Epidemiological studies usually use meta-analytical techniques (see in the same Lancet Planetary Health study used in the manuscript), which can derive summary estimates accounting for uncertainty of the single studies and different sources of heterogeneity.

Thank you for raising this point. I added significant additional explanation and analysis (pages 11-14, 34-39) to discuss how the mortality damage function is created, how uncertainty is accounted for given the limitations of the studies, and showing that it is robust across a wide variety of functional forms (also see further responses below).

When I originally did the analysis, I had run a number of these types of sensitivity checks myself, but did not include them in the paper or supplementary materials for purposes of brevity. You are right that I should have shown readers these robustness/sensitivity checks to make my results more convincing, so thank you for raising this point! The robustness check is covered on pages 36-39.

3) As a side comment, it is not clear from the text whether the author focused on one factor (temperature – as he used the Lancet Plan Health paper which only focuses on temperature), or include other stressors. For example, in line 19 page 12 “ Uncertainty is driven by uncertainty in adaptation, uncertainty in the underlying mortality-temperature relationship, and uncertainty in 20 climate model projections.” In here, one could assume that the study focuses on the direct mortality impacts attributed to temperature, but there’s no additional info before that could clarify this issue (which is crucial).

See change #1 above. Thank you for mentioning this, because I think that just focusing on temperature-related mortality makes the results of the paper clearer!

Additional comments:

The author mentioned the relevance of adaptation and its implications for mortality projections. As I understood from the text, the way the author accounted for adaptation is through the upper/lower estimates – assuming that (some of) the estimates used to derive these accounted for adaptation.

I would first ask the author to clarify this point, and then, if I’m right in my interpretation, this approach is not ideal as we would be mixing very different sources of uncertainty. It would be more appropriate to estimate the contribution of adaptation to these estimates, compare different scenarios of adaptation and discuss on the implications for policy.

Thank you for raising this, as it shows I had to better clarify the text. I provide some explanation below and added further discussion to the text in the systematic research synthesis methodology section to make this more clear, discussed in more detail below.

One of the idealized criteria that I specified for the studies used to estimate my mortality damage function is that they make their projections net of the effect of defensive adaptation. This means that, ideally, the projections made by the studies that I use should attempt to account for decisions that people make in terms of their ability to protect themselves from the mortality effects of climate change. The WHO and the Climate Impact Lab paper account for adaptation in all of their estimates including their central estimates, their high estimates, and their low estimates.

The WHO paper used the methodology developed by Yasushi Honda and colleagues to account for adaptation. The Climate impact lab also accounted for adaptation through a more sophisticated and novel method that they describe in detail in their study. Both of these studies provide uncertainty bounds (the Climate Impact Lab produced an 80% confidence interval whereas the WHO paper produced “highest” and “lowest” estimates that did not specify a specified confidence interval). These uncertainty bounds in these two studies represent uncertainty across all of the factors considered by the studies, not just adaptation.

Essentially, all I am doing is using the uncertainty as reported by the studies themselves. Because the WHO paper does not give its uncertainties statistically (i.e. no confidence intervals, just “highest” and “lowest”), I am not able to calculate precision-weighted confidence intervals as is common in other meta-analyses. The uncertainty bounds are meant to give the reader a general sense of the magnitude of the uncertainties in the mortality cost of carbon estimates (table 1) and the social cost cost of carbon estimates (table 2) (which are currently quite large). Note that this also represents an improvement over the original DICE economic damage function, as Nordhaus made no attempt to account for uncertainties, and only presented a central estimate, even though much of the literature he surveyed to construct the damage function showed that there are indeed large uncertainties in the projection of economic damages from climate change (see discussion on the last paragraph of page 12)

The Gasparrini et al paper (as you well know) did not attempt to account for adaptation in its projections. However, I still made the judgment call to include the paper despite this because most of the projections are in wealthier parts of the world. Recent studies on adaptation to mortality have suggested that wealthier countries have already largely undergone a good deal of adaptation to climate change to mitigate its mortality effects, and that much of the expected future benefits from mortality-driven adaptation can be expected to come in the developing world (Barreca et al. 2016; Carleton et al. 2019). As Reviewer 2 noted, it was somewhat of a difficult judgment call to include Gasparrini et. al 2017, in the main specification, and this is also the reason why I ran a robustness check with results that excluded the Gasparrini study.

It is also useful to emphasize that as a climate-economy integrated assessment model that is built to calculate the social cost of carbon and the mortality cost of carbon, the purpose of this analysis is simply to produce a best estimate of the expected effect of climate change on the global mortality rate in different emissions scenarios based on the projections available in the peer-reviewed literature. In order to do this accurately, such estimates ideally do their best to make this projection accounting for adaptation. In the context of this goal, it may be useful to run sensitivities to see how the social cost of carbon varies with different assumptions about different levels of adaptation. However, this is not possible with the approach that the Climate Impact Lab currently takes. They provide estimates of the increase in the expected global mortality rate in different emissions scenarios based on their best estimate of the amount of adaptation they expect to happen, which is calculated statistically in each of their 24,376 regions based on the projected income, demographics, future temperatures, and other factors in each of those regions (see their section 2 and Appendix A for details on how they do that). They do not consider higher or lower levels of adaptation but only provide their best estimate. They run uncertainties on their projection of the global mortality estimate as mentioned above, but these

uncertainties do not isolate the uncertainty in adaptation because they consider uncertainty across all of the other variables that they consider in the analysis as well, such as uncertainty in climate projections, uncertainty in the temperature-mortality relationship, and more. In future iterations of DICE-EMR, it may be possible to run social cost of carbon sensitivities across different adaptation scenarios if mortality estimates are provided for different adaptation scenarios in the projections made in future literature.

Related to the topic above, again if I understood well, the method accounts for demographic changes of the population. As acknowledged in the manuscript, the analysis was restricted by the availability of estimates by age group. The author stated that “ Because of this, the MCC captures excess deaths, not lost life years.” This is not correct, as even for the computation of excess deaths, age-specific estimates are needed if accounting for demographic changes. Otherwise, the method only accounts for the changes in the total mortality but not its distribution across ages that will change overtime in the future (see Chen et al. 2020 <https://pubmed.ncbi.nlm.nih.gov/32542573/>).

Thank you for this helpful comment and reference. I added additional text discussing this point in the last paragraph of page 13 and cite the reference you mention, and also describe further here.

Yes, DICE-EMR uses the 2019 UN Population Prospects projections for future demographic changes in its population module. Of the three studies that were used to construct the mortality damage function, the 2019 Climate Impact Lab study accounts for future demographic changes when making its projections of the future mortality rate (see sections 3, 4, and 6 of their paper) as does the 2014 WHO study in its projection of total excess deaths (see section 8.4 of their paper). The 2017 Lancet Planetary Health study, as you well know, does not, which is a further motivation to run a version of DICE-EMR that excludes the 2017 Lancet Planetary Health study in the appendix.

Ana M. Vicedo-Cabrera

Thank you, Ana, for your helpful comments and questions! Addressing them has helped to make my paper stronger and more intelligible to research-communities outside of those in economics that are not used to working with reduced-form climate-economy integrated assessment models.

Reviewer comments, second round: –

Reviewer #1 (Remarks to the Author):

Thank you for the revision, I am now happy with the manuscript.
If I could suggest two things:

1) the range for VSLY is quite high. A range from 1 to 4 would be much more reasonable, with 2 (Nordhaus' value) as the central value. I already made this remark and you responded with other estimates, but in the health literature and for policymakers used to lower values (in the UK, the official level is roughly 1), it would make more sense to focus on lower values.

2) the sensitivity analysis in table S.7 looks at a range of parameter values that is very small. For eta, I'd rather go from 0.5 to 3, and for rho, from 0 to 2.

Reviewer #2 (Remarks to the Author):

I would like to commend the author for putting substantial effort into improving the paper. I think it is much better now. The methods and assumptions are more transparent. I continue to think the exercise is valuable, though it is unfortunate that the author is so hamstrung by the inadequacies of the literature. This is the main limitation of the paper, and it still gives me some reservations, though not through any fault of the author's.

I am particularly pleased to see more details of the curve fitting. Yet, I still think even more detail may be helpful (sorry – this is a bit annoying I'm sure). Any time you are planning to present an eye popping result like this, the more transparency the better. In this case, I would say that:

- It could be helpful in Figure S2 and S3 to color code the studies
- You could consider adding the high and low estimates to the plots (in fact, as a related but separate comment, the lower estimate in Figure 5 is not discussed much at all despite being consistent with no harm/benefit from climate change. If this were an epi paper, it would not get published without reporting the upper and lower bound estimates in the abstract. I leave that to the author, but I would reflect on why you would choose not to include it.)

I'm a little confused about how cold is handled. I thought from the description on p11 that the estimates were net temperature effects, but I think the Hales paper only models heat effects.

I wonder if something about the differences between your estimate and that of Carleton et al. should be somewhere in the paper/SI along the lines of your response to my query. It is a natural comparison and an important point of departure.

Reviewer #3 (Remarks to the Author):

Thanks for the detailed answers to my comments. The manuscript is now much clearer, in particular in the description of the methodology and interpretation of the results.

I have no further comments.

Ana M. Vicedo-Cabrera

Below is R. Daniel Bressler's response to Round 2 of reviewer comments. Reviewer comments are in black, and the response is in red.

Reviewer #1 (Remarks to the Author):

Thank you for the revision, I am now happy with the manuscript. If I could suggest two things:

Thank you, Marc!

1) the range for VSLY is quite high. A range from 1 to 4 would be much more reasonable, with 2 (Nordhaus' value) as the central value. I already made this remark and you responded with other estimates, but in the health literature and for policymakers used to lower values (in the UK, the official level is roughly 1), it would make more sense to focus on lower values.

Thank you for this suggestion. I changed table 2 to match the VSLY sensitivities that you have in Table 1 of your 2019 article on *The impact of human health co-benefits on evaluations of global climate policy*, including a sensitivity where VSLY is equal to 1x consumption.

2) the sensitivity analysis in table S.7 looks at a range of parameter values that is very small. For eta, I'd rather go from 0.5 to 3, and for rho, from 0 to 2.

Thank you for this suggestion. I added a wider range of sensitivities to this table (which is now table S.8).

Reviewer #2 (Remarks to the Author):

I would like to commend the author for putting substantial effort into improving the paper. I think it is much better now. The methods and assumptions are more transparent. I continue to think the exercise is valuable, though it is unfortunate that the author is so hamstrung by the inadequacies of the literature.

Thank you!

This is the main limitation of the paper, and it still gives me some reservations, though not through any fault of the author's.

I am particularly pleased to see more details of the curve fitting. Yet, I still think even more detail may be helpful (sorry – this is a bit annoying I'm sure). Any time you are planning to present an eye popping result like this, the more transparency the better. In this case, I would say that:

- It could be helpful in Figure S2 and S3 to color code the studies

Added.

- You could consider adding the high and low estimates to the plots (in fact, as a related but separate comment, the lower estimate in Figure 5 is not discussed much at all despite being consistent with no harm/benefit from climate change. If this were an epi paper, it

would not get published without reporting the upper and lower bound estimates in the abstract. I leave that to the author, but I would reflect on why you would choose not to include it.)

Thank you for this helpful suggestion. I added in high and low estimates to the relevant charts in the paper where they were not included previously (figures 1 and 3).

I've added language to the abstract that conveys uncertainty by describing my results as central estimates and that the projection of additional 2020 emissions per excess death is in expectation. I also added an emphasis on the uncertainty as a limitation of this analysis in the mortality projections to the final conclusion paragraph of the paper, and I mentioned how future research that reduces the uncertainty around temperature-related mortality projections may increase or decrease the SCC and MCC results.

In terms of explicitly including the low and high estimates for the SCC and MCC to the abstract itself, I decided against this after much consideration. This is because, as discussed in the text, the uncertainty estimates are not actually confidence intervals because the WHO study does not present the uncertainty in its projections statistically (they give "highest" and "lowest" estimates instead of giving well-defined confidence intervals). If I were to present such estimates in the abstract, people might misinterpret these as actual confidence intervals. I tried to get around this issue by creating a few different alternative versions of the abstract that explicitly included the low and high estimates in the following ways: in version 1, I present the uncertainty in the SCC and MCC as "high" and "low" estimates. In version 2, I present the uncertainty in the SCC and MCC as ">80% confidence interval" estimates. Version 3 is the version that you see in the revision in which I say that these are central estimates and where the MCC implication is represented as a result in expectation. I shared these different versions of the abstract with a few colleagues, and the response I got was that version 1 and version 2 were unclear to them because "high" and "low" and ">80% confidence interval" are not standard ways of representing uncertainty and require significant explanation beyond the scope of what can be included in an abstract. The abstract made sense to them after reading the paper, in particular the long paragraph at the end of the systematic research synthesis subsection that discusses how uncertainty is treated in the studies used to construct the mortality damage function, and explains how the low and high mortality estimates should be interpreted. However, the abstract should be able to stand on its own in terms of clearly conveying the results. Since the standard way of conveying uncertainty is with confidence intervals, they told me that it is not clear what ">80% confidence interval" means or what "high and low" confidence intervals mean without actually reading the long paragraph in the text that explains this.

Another issue, which is also discussed in the main text, is that the original DICE model is deterministic, and the original DICE damage function doesn't attempt to account for uncertainty, so therefore the original DICE model only produces a single SCC. Whereas the DICE-EMR mortality damage function does account for uncertainty with the low and high mortality damage functions. An important finding of the paper that is presented in the abstract is showing how the SCC varies from the original DICE model when we add in the mortality damage function. It is confusing in the abstract when I present the DICE-2016 2020 SCC (which

by construction has no uncertainty), but then show the DICE-EMR 2020 SCC, which does have the low and high estimates. Of course, all of this can be explained, as I do in the paper, but the word count required to explain all of this is well beyond an appropriate word count for the abstract (to illustrate: my paragraph on uncertainty in the main text has 310 words).

Now that each of my figures show the uncertainty in the SCC and MCC estimates thanks to your helpful suggestion, my abstract refers to central estimates and expected values, and uncertainty has quite a bit of discussion in the text, I think it's likely that anyone taking even a cursory look at the paper will see that there is significant uncertainty in the SCC and MCC projections.

I'm a little confused about how cold is handled. I thought from the description on p11 that the estimates were net temperature effects, but I think the Hales paper only models heat effects.

Thank you for raising this. You are correct that the 2014 WHO paper only includes the effect of heat-related mortality in its projection whereas the Climate Impact Lab study and the Lancet Planetary Health study are net temperature-related mortality projections. The authors of the 2014 WHO study state that they make this modelling choice because the most recent IPCC report concludes that the impacts on health of more frequent heat extremes greatly outweigh the benefits of fewer cold days (on page 15 of their report): "Climate change will have some positive impacts on human health. There are likely to be reductions in cold-related mortality and morbidity in high-income populations. The most recent assessment report of the IPCC concludes, however, that the impacts on health of more frequent heat extremes greatly outweigh the benefits of fewer cold days ... The effect of cold temperatures is therefore not modelled in this assessment."

Given that this is a limitation of the WHO study relative to the idealized criteria laid out in the systematic research synthesis, however, I did an additional run of the model with an alternative specification in which the 2014 WHO report is excluded. The results of this alternative model run are shown in the new supplementary figure 5. I also added a paragraph to the main text discussing this limitation and the results of the alternative model run on page 11 (and I also added a few sentences to the end of the preceding paragraph discussing this limitation of the WHO study). As you can see, excluding the WHO study from the mortality damage function has a very minor effect on the results, as the SCC increases from \$258 to \$264 and the MCC increases from 2.26×10^{-4} to 2.38×10^{-4} .

I wonder if something about the differences between your estimate and that of Carleton et al. should be somewhere in the paper/SI along the lines of your response to my query. It is a natural comparison and an important point of departure.

This is a very good point. I added my response to your initial question in your first referee report about the DICE-EMR results vs. the Carleton et al. results to its own paragraph on the bottom of page 19.

Reviewer #3 (Remarks to the Author):

Thanks for the detailed answers to my comments. The manuscript is now much clearer, in particular in the description of the methodology and interpretation of the results. I have no further comments. Ana M. Vicedo-Cabrera

Thank you, Ana!